# Global geo-hazard risk assessment of long-span bridges enhanced with InSAR availability

Dominika Malinowska [1,2] ✉, Pietro Milillo [3,4,5], Cormac Reale [2], Chris Blenkinsopp [2] & Giorgia Giardina [1]

While a geo-hazard risk assessment of bridges is crucial for achieving the United Nations' Sustainable Development Goals, state-of-the-art methods for evaluation of risk neglect the temporal dimension of structural vulnerability, overlooking how monitoring systems like Structural Health Monitoring sensors and Multi-Temporal Interferometric Synthetic Aperture Radar can continuously track bridge conditions. Moreover, despite Structural Health Monitoring systems being sparsely installed, no research has quantified the global potential of this spaceborne radar-based technique as a complementary monitoring solution for bridges. This study introduces a method that integrates monitoring availability into structural vulnerability assessments and evaluates the global risk of long-span bridges affected by subsidence and landslides. Findings revealed that while fewer than 20% of bridges have Structural Health Monitoring systems, spaceborne monitoring could provide monitoring for over 60% of structures, leveraging Sentinel-1's global coverage. Incorporating this satellite remote sensing approach into routine assessments could decrease the number of bridges classified as high-risk by one-third. Moreover, half of the remaining high-risk structures could benefit from spaceborne monitoring, highlighting the technique's potential to enhance structural safety and resilience, especially in economically disadvantaged regions.

Bridges play a critical role within transportation networks, facilitating regional connections and providing access to work, education, healthcare, and emergency services. Despite their importance, bridges are among the most vulnerable components of these networks, regularly impacted by natural hazards whose frequency is expected to increase further due to climate change effects[1,2]. Therefore, enhancing their reliability is crucial, particularly to meet the United Nations' (UN) sustainable development goals (SDG) 9, 10, and 11, focusing on building resilient infrastructure, especially in developing countries,

reducing inequalities, and improving the safety of built environments[3]. To ensure that infrastructure, including bridges, is safeguarded against hazards, the Sendai Framework for Disaster Risk Reduction calls for innovative multi-hazard risk assessment methods and improved utilisation of in situ and remotely sensed information, including space-based technologies, to enhance monitoring and communication of disaster risks[4]. Consequently, there is a need for novel risk assessment approaches that leverage these technologies to improve the resilience and safety of bridge infrastructure.

[1]Department of Geoscience & Engineering, Delft University of Technology, Delft, The Netherlands. [2]Department of Architecture and Civil Engineering, University of Bath, Bath, UK. [3]Department of Civil and Environmental Engineering, Cullen College of Engineering, University of Houston, Houston, TX, USA. [4]Department of Earth and Atmospheric Science, Cullen College of Engineering, University of Houston, Houston, TX, USA. [5]Microwaves and Radar Institute, German Aerospace Center, Weßling, Germany. ✉e-mail: dum22@bath.ac.uk

Risk in the context of infrastructure management is defined as the potential for future events to result in negative consequences for valuable assets[5]. Assessment methodologies span from simple approaches that evaluate individual risk scenarios, through methods using comparative scoring systems, to complex probabilistic techniques, with more comprehensive approaches requiring detailed inputs that make application at larger scales challenging[6]. Moreover, transportation network risk modelling lacks standardisation[7]. Nevertheless, in civil engineering, risk is commonly calculated using the risk triangle approach, where it is the product of hazard, exposure, and vulnerability[5,8,9]. Hazards are typically defined as sources of threats that can cause harm to assets[5,10]. Those affecting bridges include both natural (floods, earthquakes, landslides) and anthropogenic (overloading, accidents, negligence) threats[11], with hydraulic-related issues being particularly prevalent causes of collapse[12–14]. While asset management has historically focused on overloading and routine weather events[7], climate change is expected to significantly increase hazard exposure through more extreme events[2], necessitating broader hazard assessment considering natural hazards in bridge management[15].

Definitions of exposure and vulnerability vary depending on the context[9,10]. Exposure may refer to people or structures at risk, the value of assets vulnerable to hazards, or be quantified through population density or the types of assets in an area[9,16]. Exposure assessment methods include critical infrastructure mapping for large-scale risk assessments[17] and road typology for network analyses[18], all ultimately evaluating potential community impact. In contrast, vulnerability may be measured as expected loss upon asset failure[5], characteristics making structures susceptible to destruction[19], or factors increasing hazard susceptibility[9,20]. Despite these varying definitions, they all characterise vulnerability as a measure of how significantly a hazard affects a system. Sometimes, the same indicators, such as the density of the built environment, may be used to assess both exposure and vulnerability, highlighting the conceptual overlap between these risk components[21]. For bridges, structural vulnerability is critical as it is time-variant, requiring regular monitoring and updates[22].

Regular visual inspections are crucial for monitoring structural vulnerability and identifying early signs of deterioration in bridges[22]. However, these in-person inspections present several limitations: (1) they are costly and often subjective, with assigned scores varying significantly between inspectors[23,24], (2) visual inspections may miss early signs of deterioration or fail to detect changes that develop between the typical 2-year inspection intervals, making traditional periodic approaches insufficient for timely detection of structural issues[25], (3) and access difficulties can reduce inspection frequency, particularly when inspection needs were not considered during design[26]. Structural health monitoring (SHM) sensors offer a more cost-effective solution for real-time monitoring, reducing labour and material costs[27] while providing early warnings of potential damage[28]. Although SHM cannot entirely replace in-person inspections, it serves as a valuable complementary tool, helping identify specific elements requiring closer inspection[26]. Despite these advantages, SHM implementation remains limited primarily to newer bridges and specific concern cases[29], with global data indicating installation on only a small subset of bridges[30]. Furthermore, proactive installation on existing bridges is constrained by high costs and operational complexities of data management[29].

Remote sensing offers an alternative to SHM sensors and can support visual inspections, particularly when direct access to a structure is challenging[31–33]. Notably, spaceborne monitoring using synthetic aperture radar (SAR) offers frequently acquired, high-resolution imagery with global coverage and extensive historical archives[20]. For bridges specifically, multi-temporal interferometric SAR (MT-InSAR) can complement traditional inspections[31,34–36] by processing interferogram stacks to detect temporal displacement trends[37,38]. Employing pixels with stable scattering properties known as persistent

scatterers (PSs), MT-InSAR enables precise displacement measurements and is particularly effective for tracking slow-moving phenomena[39,40], including bridge displacement caused by slow-moving landslides and subsidence[41–43], and for detecting anomalies across extensive regions[44–46]. The European satellite Sentinel-1 provides free global SAR coverage, theoretically enabling worldwide structural monitoring. However, some bridges may not interact effectively with radar wavelengths, limiting MT-InSAR's applicability. Previously, assessing the feasibility of a structure's spaceborne monitoring required data preprocessing, but recent advances enable the prediction of MT-InSAR PS point availability[47], allowing for assessment of MT-InSAR usability for bridge monitoring prior to data acquisition.

Risk assessment is crucial for identifying vulnerable bridges and prioritising maintenance, but comprehensive frameworks must consider multiple hazards, structural conditions, strategic importance, and inherent uncertainties[1,11,15,48]. Despite increased research on modelling natural hazard effects on transportation networks, there remains a gap in integrating risk-based insights into decision-making frameworks[7]. While some approaches use risk analysis to identify critical elements for monitoring or to prioritise high-probability damage areas for inspection and maintenance[29,49], these primarily focus on where traditional SHM systems should be installed, neglecting the integration of remote sensing technologies with risk frameworks. Several methods have been proposed for considering spaceborne radar data in risk assessment: displacement data can be combined with ground-based measurements for enhanced monitoring[50], with exposure metrics to prioritise critical bridges[18], or with natural hazard information to measure hazard-induced deformation[51], directly as a measure of a hazard[52–54], to assess infrastructure vulnerability[55], or to interpret MT-InSAR results[56,57]. However, the integration of MT-InSAR monitoring availability into risk frameworks remains unexplored.

Monitoring is crucial in the context of landslides and ground subsidence, which present unique challenges for bridge management due to their progressive nature and the complexity of their structural response. Landslides cause foundation and abutment sliding and undercutting, impact piers, and can even affect overall structural integrity[58,59]. Ground subsidence induces differential settlements that cause bridge displacement in horizontal and vertical dimensions, leading to structural tilting and torsion, and causing visible damage, including abutment deformation, deck cracking, and span rotational twisting[60]. The structural response involves complex stress redistribution where the superstructure must accommodate both subsidence-induced stresses and normal loads amongst misaligned supports, with time-dependent interactions between ongoing ground movement and bridge material behaviours exacerbating structural deterioration and potentially causing impact damage between bridge elements[61]. These geo-hazards can exhibit coupled effects with seismic activity, where pre-existing ground displacement or landslides amplify seismic response and increase overall structural vulnerability[59,61,62]. The impact of localised displacement caused by geo-hazards on bridges' structural health, such as that affecting piers, can be more difficult to detect during periodic inspections compared to the effect of more global displacement that affects the whole structure and often results in more visible functionality issues[61]. Thus, remote sensing techniques are particularly valuable for geo-hazard-affected bridges, as they could detect localised deformation patterns that might be overlooked in routine inspections and enable frequent monitoring of progressive movements.

Therefore, the research gap in MT-InSAR-based monitoring of geo-hazard-affected bridges is two-fold. Firstly, whilst MT-InSAR is valuable for monitoring such bridges, PS availability is not guaranteed and could limit wide-scale application. We therefore employ recent advances in PS prediction to quantify MT-InSAR's potential for bridge monitoring. Secondly, geo-hazard-affected bridges exhibit time-variant vulnerability through localised dynamic changes that

traditional inspections may miss. SHM and MT-InSAR provide more frequent status updates than visual inspections, reducing uncertainties. We thus propose integrating monitoring system availability into geo-hazard risk assessments of bridges.

Our approach quantifies structural vulnerability in two stages: first, by evaluating the bridge's physical characteristics, and then by integrating monitoring system availability as a temporal vulnerability factor, assuming that continuous monitoring reduces inherent uncertainties regarding the current structural health of the structure. We evaluate MT-InSAR capability for monitoring bridges with Sentinel-1 data through the prediction of measurement point density combined with spacecraft coverage data, while SHM availability is determined from the existing database. Applied to a global dataset of 744 long-span bridges, i.e. those with main spans exceeding 150 m, our methodology employs open-source subsidence and landslide hazards global datasets, calculates exposure metrics based on bridge functional characteristics, and combines these with the enhanced vulnerability factors to generate comprehensive risk profiles that inform monitoring decisions.

Through worldwide analysis of spaceborne monitoring availability, our results highlight a significant monitoring gap, demonstrating that SHM systems are present on relatively few bridges, whereas MT-InSAR via Sentinel-1 could provide observation capabilities for a much greater proportion of structures. Including this expanded monitoring capability in the risk assessment framework refines the risk classification methodology and demonstrates how incorporating spaceborne monitoring can improve risk level definitions and reduce the number of bridges classified as high-risk. Importantly, many remaining high-risk structures could benefit from spaceborne monitoring, highlighting MT-InSAR's potential to enhance bridge safety and resilience worldwide, particularly in economically disadvantaged regions. The proposed approach provides a means of leveraging remote sensing suitability to prioritise bridges for monitoring and create a more dynamic risk assessment that accounts for the temporal dimension of structural vulnerability. Therefore, it can serve as actionable guidance for stakeholders on optimising the deployment of remote sensing technologies, SHM installations, and in-person inspection schedules based on risk-informed decision frameworks.

## Results

This study investigates the global risks affecting long-span bridges by conducting a risk assessment that considers monitoring availability in the evaluation of structural vulnerability. Using this approach, we performed a comprehensive geo-hazard risk assessment for bridges catalogued in the long-span bridges database[30]. A detailed explanation of the risk calculation methodology is provided in the Methods section.

### PS availability and spaceborne monitoring

Each bridge was assigned a PS availability class based on two predicted metrics: average PS density per 100 m and the proportion of the bridge covered by PS, following Table 1. As illustrated in Fig. 1a, over half of the bridges exhibited high or very high PS availability, indicating their potential suitability for MT-InSAR monitoring. However, the temporal frequency of Sentinel-1 data acquisition and the availability of ascending and descending datasets varied significantly among bridges. Only 31% of the bridges recorded in the database were covered by both tracks every 6 days, while almost 40% had data available from only one flight direction every 12 days (see Fig. 1b).

The combined assessment of PS and Sentinel-1 data availability was used to derive the spaceborne monitoring class. The limited Sentinel-1 data availability caused some bridges with potentially high PS populations to be assigned to lower spaceborne monitoring classes. Consequently, as shown in Fig. 1c, only 21% of the bridges were classified as having very high spaceborne monitoring capabilities. Nevertheless, approximately 60% of the long-span bridges in the database

### Table 1 | PS availability classification categories

| % pixels with ≥1 PS | #PS per 100 m | | | | |
|---|---|---|---|---|---|
| | <1 | [1,3) | [3,5) | [5,10) | ≥10 |
| ≥80% | 0.4 | 0.6 | 0.8 | 1 | 1 |
| [60–80%) | 0.2 | 0.4 | 0.6 | 0.8 | 1 |
| [40–60%) | 0.2 | 0.4 | 0.4 | 0.6 | 0.8 |
| [20%,40%) | 0.2 | 0.2 | 0.4 | 0.4 | 0.6 |
| <20% | 0.2 | 0.2 | 0.2 | 0.2 | 0.4 |

PS availability was categorised based on two criteria: the proportion of pixels containing at least one PS and the average number of PSs per 100 m. The classification assigns values to each class as follows: very low—[0, 0.2], low—(0.2, 0.4], medium—(0.4, 0.6], high—(0.6, 0.8], and very high—(0.8, 1.0].

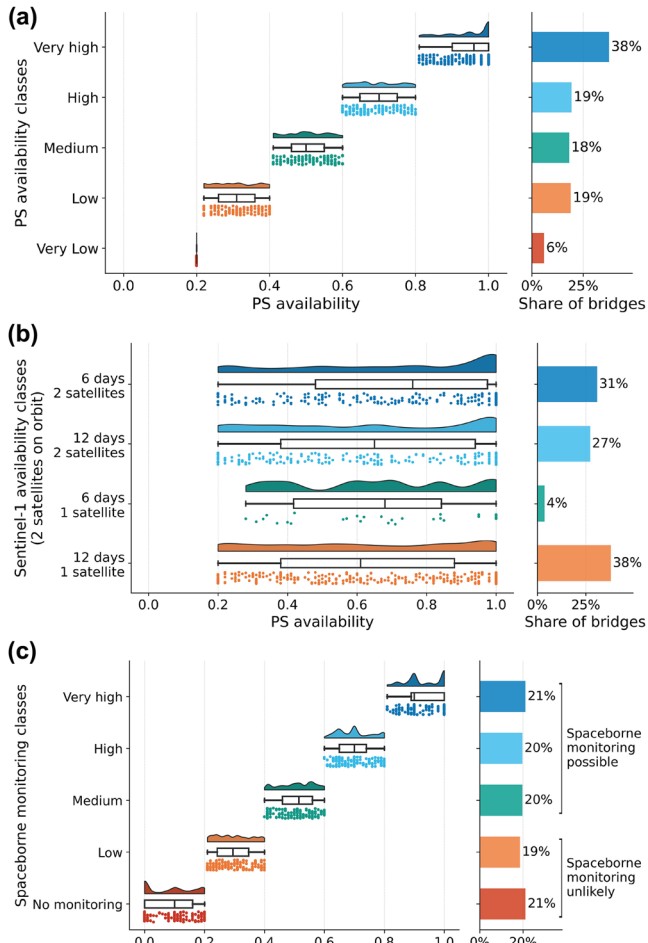

Fig. 1 | Comparative histograms of bridge monitoring capabilities. a PS availability over bridges, b Sentinel-1 satellite data coverage, and c effective spaceborne monitoring potential, calculated as PS availability adjusted for Sentinel-1 data availability. Source data are provided as a Source data file.

were classified as medium or higher in terms of spaceborne monitoring potential.

### PS availability and bridge physical properties

The bridge database used in this study specifies the materials of the bridge's deck, piers or pylons, and cables or trusses. To simplify the analysis of PS availability correlation with material, each bridge was categorised into one of four material groups: (1) steel if both deck and piers/pylons were steel; (2) concrete if both were concrete; (3) composite if the deck was steel and piers/pylons were concrete; and (4)

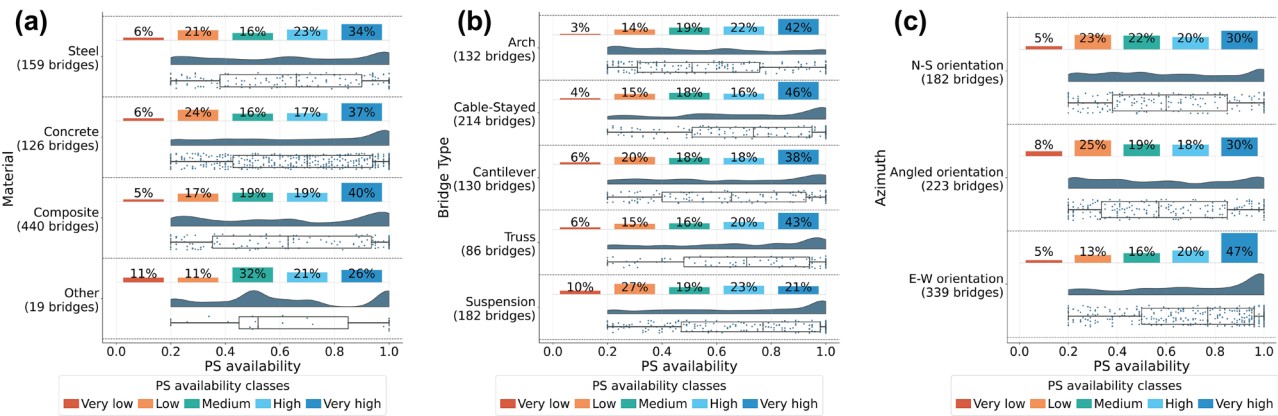

**Fig. 2 | Distribution of PS availability depending on different bridge characteristics. a** Distribution of PS availability by the material it is made from, **b** distribution of PS availability depending on bridge type, and **c** distribution of PS availability by azimuth of the bridge, classified into N-S (0–30°, 150–210°, 330–360°), Angled (30–60°, 120–150°, 210–240°, 300–330°), and E-W orientations (60–120°, 240–300°). Source data are provided as a Source data file.

other, for all remaining combinations. Figure 2a illustrates the relationship between bridge material and PS availability. Interestingly, no clear correlation was observed.

When considering the bridge type, as shown in Fig. 2b, suspension bridges appeared to be the least favourable for space-based observation. Other bridge types demonstrated similarly good monitoring capabilities. This analysis is further extended in Supplementary Fig. 1, which plots PS availability by segment for each bridge type. Each bridge was divided into five equal-length segments: two edge sections, two intermediate sections, and a central span. The mean of the two edges and the two intermediate segments for a given bridge were used for the plot. For all bridge typologies, the central span exhibits the worst monitoring capabilities. This difference is particularly prominent in suspension bridges, although their edge spans are relatively well monitored.

SAR satellites follow polar orbits, moving approximately in the North-South direction. Figure 2c presents the influence of bridge orientation on the availability of PSs. Bridges perpendicular to the SAR orbit orientation have the highest proportion of very high PS availability. In contrast, those closer to the N-S direction have a larger share of bridges that cannot be monitored. However, a third of those bridges that are almost parallel to the N-S orientation still provided very strong reflections.

## Comparison of SHM and spaceborne monitoring

While Sentinel-1 data are available worldwide, spaceborne monitoring capabilities vary depending on bridge properties. Conversely, SHM sensors are only installed on specific bridges, predominately those recently built or with known structural issues. To facilitate comparison between in-situ and space-based sensors, spaceborne monitoring was divided into two categories: 'unlikely' if monitoring was previously classified as none or low, and 'possible' if it was medium, high, or very high.

Figure 3a illustrates the spatial distribution of monitoring availability by method, highlighting that satellite monitoring provides more balanced capabilities worldwide than the SHM. The binary availability of SHM sensors leaves over 80% of bridges globally without monitoring. In contrast, space-based monitoring offers some level of observation for 61% of bridges. While SHM sensors are primarily installed in Asia, Europe, and the Middle East, spaceborne monitoring also provides observations in other areas. This is particularly evident in regions such as Africa and Oceania, where almost no long-span bridges have SHM sensors, but well over half could be monitored from space.

Interestingly, when considering both currently installed SHM sensors and potential spaceborne monitoring together (Fig. 3b), only 119 bridges are left without any monitoring available, and 376, around

half of the whole dataset, of those without SHM sensors have a medium or higher spaceborne monitoring category, making them potentially observable with SAR.

## Structural vulnerability and monitoring

Figure 4a reveals that North America has long-span bridges in the worst structural condition, with almost 70% in the two highest vulnerability classes. Africa follows closely, with almost half of its bridges having high or very high structural vulnerability, and Europe, where close to 40% fall into these categories. Conversely, Middle Eastern bridges exhibit the best structural condition.

The comparison of SHM and spaceborne monitoring capabilities against structural vulnerability (Fig. 4b) highlights that while there is no dependency between vulnerability and monitoring capabilities for either method (i.e. structurally vulnerable bridges are not prioritised for monitoring), remote sensing monitoring usually provides very high monitoring for more than the proportion of bridges with SHM. Additionally, it allows for monitoring with lower capabilities for many more bridges, leaving fewer bridges entirely without monitoring.

## Including SHM and spaceborne monitoring availability in the risk assessment

The distribution of geo-hazard risk that includes SHM availability (Fig. 5a) revealed that over 20% of long-span bridges globally are at very high risk, followed by more than 50% in the high-risk category. Regional analysis shows North America has the highest share of bridges at very high and high risk, while the Middle East has the highest number globally in the low-risk category. This can be related to the structural vulnerability as previously presented in Fig. 4a.

As shown in Fig. 5b, when only SHM was included in the monitoring factor, there were over 150 bridges at very high risk, with only a small portion of these structures having a monitoring system available. Incorporating satellite monitoring capabilities into the risk analysis reduces the epistemic uncertainty, leading to a reduction in risk for many structures compared to when only SHM was considered, enabling asset managers to refine risk registers over time. In total, over 50 bridges had their risk changed from very high to high. For those that remained in the very high category, about half could be monitored with either in-situ sensors or space-based data.

Figure 6 illustrates the spatial distribution of risk changes when comparing risk assessment considering only SHM sensors to that including both in-situ sensors and spaceborne monitoring. Globally, employing MT-InSAR for routine bridge monitoring could decrease the mean regional risk by over 4%, with the highest reduction observed in Africa, Europe, and the Middle East.

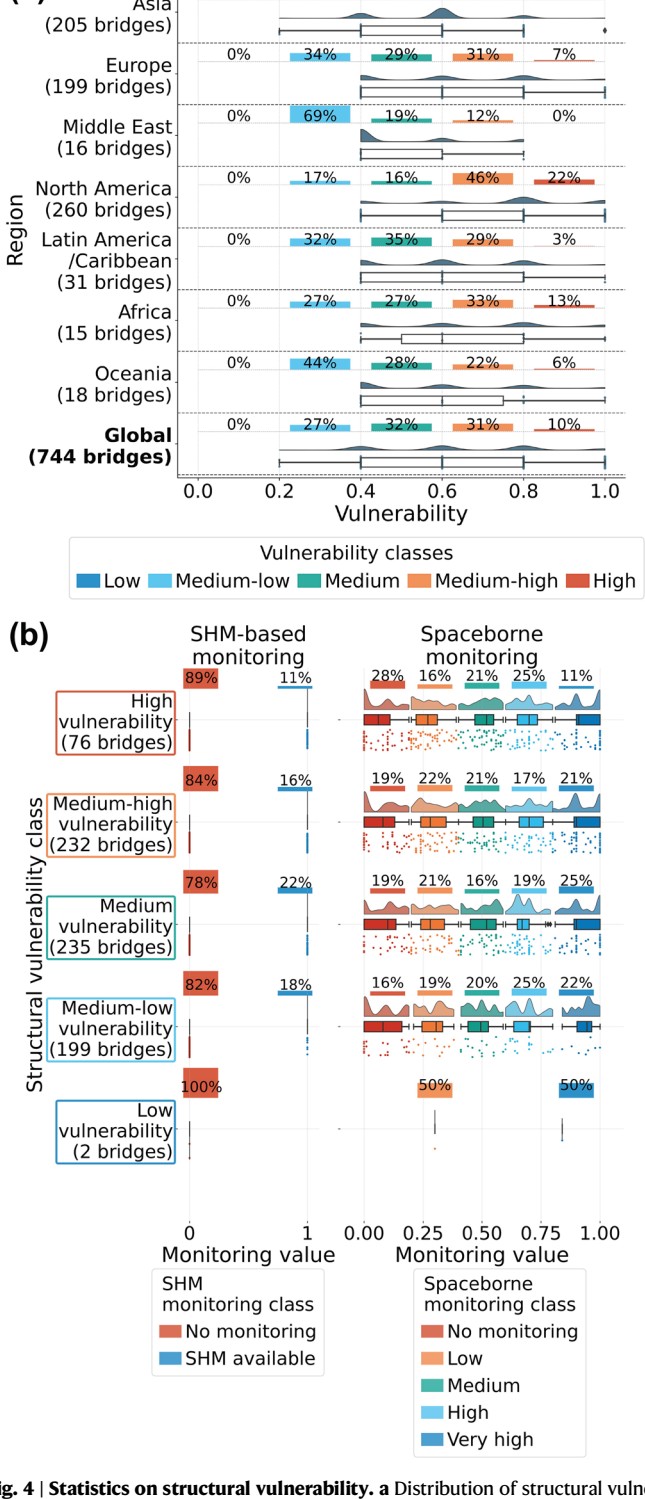

**Fig. 3 | Comparison of SHM and spaceborne monitoring capabilities.**
**a** Distribution of SHM and spaceborne monitoring capabilities by region. SHM availability is represented as a binary variable (0 = no sensors installed, 1 = sensors available); spaceborne monitoring capability ranges from 0 to 1, with bridges scoring ≥0.4 considered observable via satellite. **b** Overlap between bridges currently monitored with SHM systems and those for which remote sensing could provide monitoring. Numbers on the side indicate the count of bridges in the considered category. Source data are provided as a Source data file.

**Fig. 4 | Statistics on structural vulnerability. a** Distribution of structural vulnerability by region. **b** Comparison between the share of bridges with SHM installed and the bridges possible to monitor with Sentinel-1, depending on the bridge's structural vulnerability. Source data are provided as a Source data file.

## Discussion

This article presents a global bridge geo-hazard risk assessment methodology that considers the availability of both SHM sensors and spaceborne monitoring. The analysis encompassed subsidence and landslide hazards, exposure, and structural vulnerability. Furthermore,

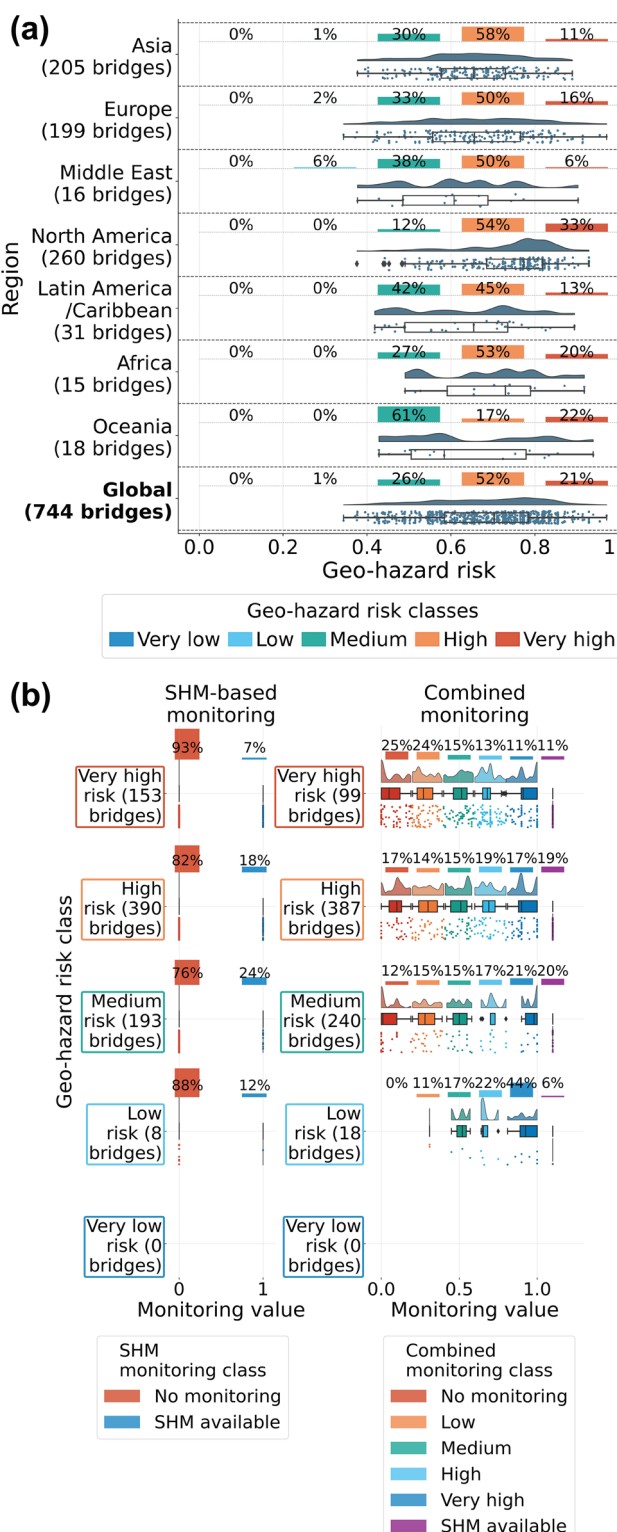

**Fig. 5 | Distribution of geo-risk hazard spatially and by class. a** Spatial distribution of geo-hazard risk, including only SHM monitoring capabilities in the calculation. **b** Distribution of geo-hazard risk with monitoring capabilities: SHM only (left) and combined SHM-spaceborne monitoring (right). Source data are provided as a Source data file.

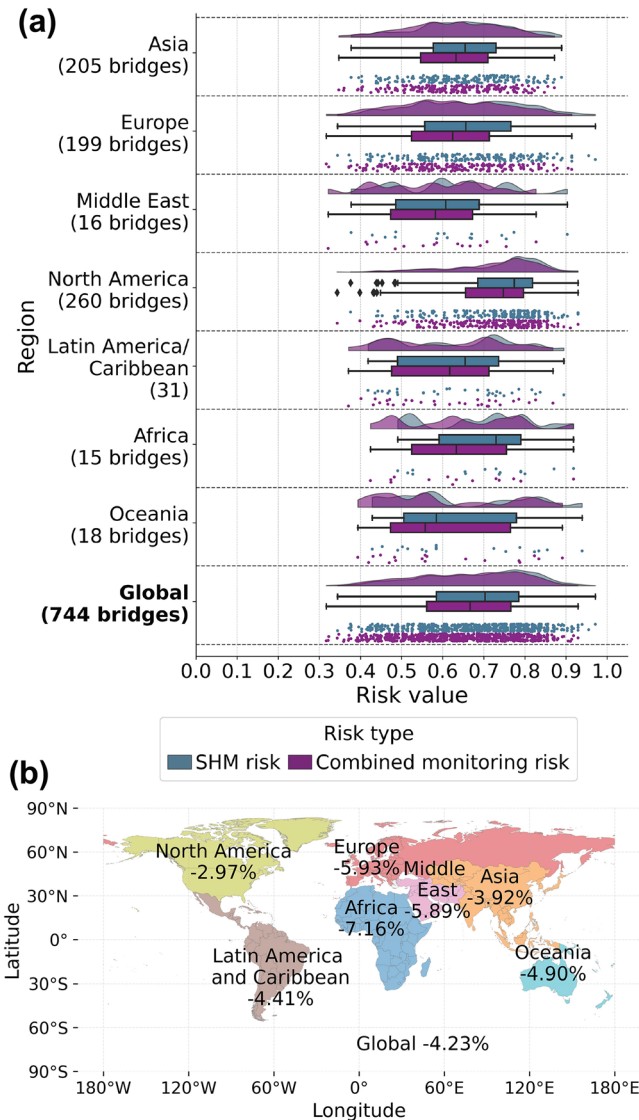

**Fig. 6 | Changes in risk by region. a** Distribution of regional risk with and without spaceborne monitoring included in the assessment. **b** Relative changes in mean risk when spaceborne monitoring is included in the assessment, grouped by region. Source data are provided as a Source data file.

a method was introduced to assess a structure's suitability for MT-InSAR monitoring, which accounts for monitoring availability when determining structural vulnerability. The methodology was applied to a long-span bridge database to evaluate these structures' suitability for space-based monitoring and provide a worldwide risk assessment. Additionally, the results present factors affecting a structure's suitability for spaceborne monitoring and highlight the potential of MT-InSAR as a complementary methodology for structural evaluation due to its worldwide availability. Incorporating PS data into risk assessments provides more accurate risk registers through epistemic uncertainty reduction, enabling better risk prioritisation and maintenance planning.

Analysis of PSs availability revealed that over half of the bridges recorded in the database were densely populated with PSs. However, the effective spaceborne monitoring potential decreased when Sentinel-1 data availability was factored into the analysis. This underscores the crucial need for the worldwide availability of frequent acquisitions from ascending and descending tracks to analyse bridge structural health comprehensively. Notably, the availability of Sentinel-1 data decreased a few years ago due to the failure of one of the satellites. The Sentinel-1 mission originally consisted of two spacecraft, Sentinel-1A and -1B, following the same orbit but phased to provide global coverage with a 6-day repeat time in Europe and a 12-day repeat time elsewhere. Furthermore, observation scenarios were planned to

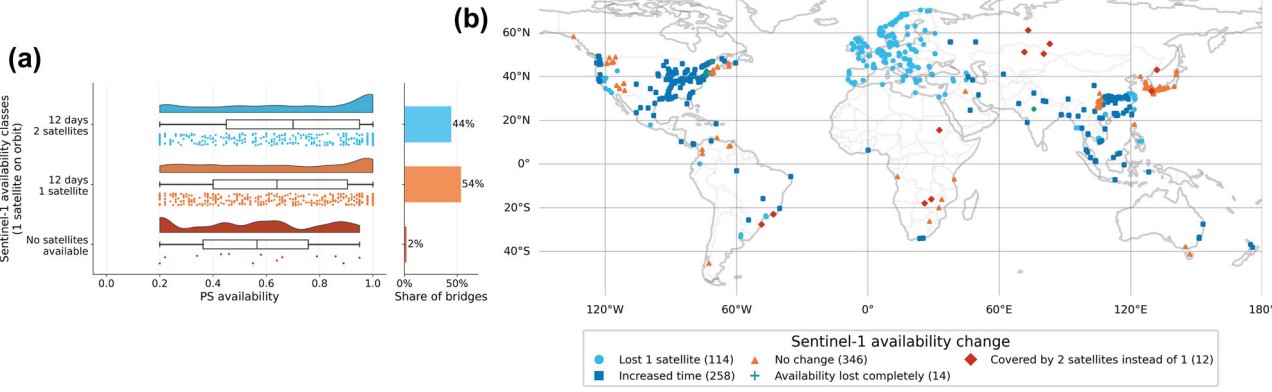

**Fig. 7 | Changes in Sentinel-1 availability due to the failure of one of the satellites. a** Comparison of Sentinel-1 availability before and after Sentinel-1B failure. **b** Map showing how the availability of Sentinel-1 changed after one of the satellites was lost. Source data are provided as a Source data file.

provide both ascending and descending flight directions, where possible, enabling the decomposition of MT-InSAR measurements into vertical and east-west components, which is crucial for appropriate structural health analysis. However, Sentinel-1B was damaged and lost in 2022, significantly decreasing data availability until its replacement, Sentinel-1C, was launched in December 2024[63]. Figure 7a illustrates how the failure of one satellite eliminated the possibility of providing data every 6 days, reduced the number of bridges with both flight directions available, and left some structures without any spaceborne monitoring. The reduced satellite capacity created regional disparities in monitoring availability. Highly developed regions such as Europe maintained data from two flight directions, although with longer revisit times. However, bridges in less privileged areas experienced significantly reduced capabilities, with some in Africa and Latin America losing spaceborne monitoring entirely (see Fig. 7b). With the launch of the Sentinel-1C, one can expect the return of more comprehensive monitoring capabilities. However, analysis dependent on temporal datasets will be hindered for the near term due to the years of reduced coverage.

Correlations between bridge PS availability and bridge characteristics were analysed. Whilst the physical properties of materials can impact the strength of the SAR signal return, no clear correlation between the bridge material and its potential for monitoring was found. This could be attributed to the fact that bridges are typically constructed from steel and concrete, both of which are characterised by good reflective properties, or it might be because the radar signal is primarily reflected by the superstructure or additional equipment installed on the bridge, such as signs or lampposts. Analysis of bridge type against PS availability revealed significant differences between typologies. Suspension bridges were found to be the least suitable targets for spaceborne monitoring, while for other types, the difference in PS availability classes was negligible. This is due to suspension bridges being more affected by vibrations, especially in the central span, that can cause loss of coherence and consequently decreased availability of PS[64]. Although these bridges have fewer points overall, edge sections are monitored relatively well, suggesting that MT-InSAR could still provide meaningful information about the structural health of bridge piers and abutments. However, the limited availability of PSs in the central span represents a limitation of MT-InSAR usability for bridge monitoring. The bridge's azimuth did not significantly affect PS availability. Although previous literature demonstrated that linear infrastructure, such as roads, has better reflectivity when perpendicular to the satellite orbit[34], the analysis performed in this study showed that this effect is not pronounced for long-span bridges. This may be attributed to Sentinel-1's asymmetric spatial resolution, which is approximately 5 m in the range direction (roughly E-W) and 20 m in the azimuth direction (roughly N-S). Consequently, E-W oriented

bridges are sampled by more pixels along their length compared to N-S oriented structures, potentially providing more opportunities for PS detection. While E-W-oriented bridges are favourable, observing bridges with different azimuths is also feasible. However, it should be noted that the analysis of movement along the length of a bridge with azimuth parallel to N-S (e.g. thermal expansion) would be restricted due to MT-InSAR limitations in measuring displacements along the N-S direction. Nevertheless, spaceborne measurements would still be capable of detecting vertical deformations on such bridges, which could provide some insights into structural health.

The comparison between SHM- and space-based monitoring revealed that whilst SHM is installed on only about 20% of all long-span bridges in the database, spaceborne monitoring provides a reliable level of observation for more than 60% of the database. Moreover, when both monitoring methods are considered together, the capabilities are further enhanced as some bridges that cannot be monitored from space have SHM systems already installed, and some have both monitoring systems available, providing stakeholders with significantly improved confidence regarding bridge structural conditions. These in-situ sensors are primarily installed in developed regions such as Europe, Asia, and the Middle East, possibly due to the high cost of SHM installation and operation. At the same time, spaceborne monitoring provides more democratic capabilities, allowing for monitoring a significant share of the bridges in Africa, Oceania, Latin America, and the Caribbean, which have almost no SHM sensors. This underscores the global potential of MT-InSAR monitoring as a method capable of providing information about the structural health of bridges worldwide, hence providing a means to enhance infrastructure resilience and meet SDGs.

The analysis indicated that North American bridges are in the worst structural condition. This correlates with the age of the bridges, as there was a peak in North American bridge construction in the 1960s, meaning many of these bridges are near or beyond their design lives[30]. African bridges were also found to be in poor structural condition and simultaneously lacked any SHM sensors. This demonstrates that, thus far, structural vulnerability has not been a primary consideration when selecting bridges for SHM sensor installation. This issue could be addressed by employing MT-InSAR, which provides data for 67% of African bridges, including data archives that could be utilised for retrospective analysis of temporal trends. However, the recent decrease in Sentinel-1 availability restricts this analysis, once again highlighting the importance of providing SAR data globally. Notably, spaceborne monitoring could provide information for most highly structurally vulnerable bridges, compared to only a small portion with available SHM sensors. Thus, employing MT-InSAR to monitor bridges globally would enhance knowledge about the current structural health of these structures, allowing those with deteriorating

structural conditions to be addressed first and, in turn, addressing SDG 9 by improving city resilience.

Risk was calculated in two ways: considering structural vulnerability with only SHM availability and including the availability of both in-situ sensors and spaceborne monitoring. Analysis of the former showed that North American and African bridges are at higher risk, which could be correlated to their poor structural condition. When the latter, more comprehensive monitoring was examined, the number of bridges at very high and high risk was significantly reduced. A third of structures in the very high-risk category had their risk reduced to a lower class, with about half of the structures left being potentially monitored through either of the considered technologies. Importantly, when considering both monitoring methods, bridges that already had SHM systems installed maintained their existing risk level, while structures without SHM that could be monitored from space experienced risk reductions due to decreased uncertainties in structural condition assessment through additional spaceborne data points. The spatial analysis revealed a global average risk reduction of over 4%, with particularly notable decreases in Africa, Europe, and the Middle East. In Africa, this is likely due to poor bridge conditions and limited SHM availability; in Europe, high structural vulnerability paired with excellent Sentinel-1 coverage played a role; and in the Middle East, dense PS availability, thanks to favourable reflective properties of the bridges and good spaceborne data availability, contributed to the reduction. Nevertheless, the reduction was also observable in other areas of the world. Notably, including spaceborne monitoring significantly increased the share of monitored bridges in each risk category. Consequently, incorporating information about monitoring availability in risk analysis can help identify bridges that are affected by hazards, in poor structural conditions, and would cause significant disruption in case of failures. Such structures can then be prioritised for more frequent in-person inspections, new SHM system installation, or MT-InSAR monitoring.

The methodology presented in this article provides a degree of flexibility so that stakeholders who wish to use it to prioritise their inspection resource use can, for example, add more accurate information about structural health or sensor availability, include other hazards, use different metrics for exposure assessment, or change the maximum value of the monitoring factor. The monitoring factor used in structural vulnerability calculations reflects the assumption that the availability of remotely sensed data allows more frequent structural health updates, thereby reducing uncertainty in the inherently time-variant structural vulnerability and consequently decreasing overall bridge risk. Monitoring improves knowledge about the structure; hence, we adopted a coefficient value of 1.35, commonly used in building codes and standards, such as Eurocode-8[65] and FEMA 356[66], to account for uncertainties associated with (lack of) knowledge about building materials. Although this is a standardised value, it has not previously been applied in this context. Therefore, to fully understand the impact of this value selection on the final risk score, we performed a sensitivity analysis. Supplementary Fig. 5 shows that changes due to different monitoring factor selections would be minor, with few bridges changing their assigned risk category. Additionally, whilst this work assumes that monitoring availability reduces epistemic uncertainties in time-variant structural vulnerability, it does not explicitly quantify uncertainty propagation through the risk model. Future work could therefore investigate more comprehensive quantification of how monitoring availability impacts vulnerability assessment, including the implications of MT-InSAR measurement limitations and potential synergistic effects when both SHM and spaceborne monitoring are available. Such developments would enhance stakeholder confidence in risk scores and improve the robustness of prioritisation decisions.

In this paper, the analysis considered two hazards to demonstrate how multi-hazard assessment can be included. However, it should be noted that the interdependence between these was not considered. Whilst considering the interdependencies between hazards can improve accuracy and lead to more informed decisions, it comes at the cost of more complicated modelling and higher requirements for information, making it extremely difficult to implement on a global scale[67]. However, if the method were to be applied on a more regional scale, including the cascading effect of multiple hazards could improve the accuracy of the assessment. Moreover, the geo-hazard assessment could include the temporal scale of hazards. Whilst both hazards considered in this article have a temporal scale that ranges from seconds to days (for landslides) or even years (for subsidence), other hazards that can impact bridges, such as earthquakes, can have a much shorter temporal scale[68]. In such cases, monitoring might not provide timely warnings about a change in structural health, and further research could explore how the geo-hazard analysis might be extended to account for this aspect.

Several limitations of the methodology stemming from the accuracy of the input data should be acknowledged. Bridge line identification was automated from OSM data and may be subject to bias, as OSM is not fully complete and sometimes inaccurate[69,70]. Due to data collection challenges encountered by the authors of the bridge database[30], certain regions, particularly Africa and Oceania, are represented by fewer bridges in the dataset. This spatial imbalance potentially introduces bias in regional statistics, which should therefore be interpreted with caution. Furthermore, the exposure and vulnerability assessments necessarily rely on globally available datasets that are inherently static and of relatively coarse resolution. This data limitation has several implications for the risk assessment: regional variations in construction standards and engineering practices may not be adequately captured by broad typological categories, whilst static datasets cannot account for temporal changes in bridge conditions. These limitations, i.e. spatial imbalance in the bridge database and coarse resolution of data underlying exposure and vulnerability evaluation, reflect the inherent trade-off in conducting global-scale infrastructure analysis, where comprehensive spatial coverage must be balanced against data granularity and completeness. Nevertheless, the methodology presented here provides a valuable framework that can be readily applied to more comprehensive regional datasets as they become available. Moreover, the bridge database used in the study does not specify SHM sensor types, so our assumption of 'ideal' monitoring may overestimate its capabilities. Additionally, MT-InSAR has inherent limitations requiring expert interpretation to distinguish expected movements (e.g. thermal dilation) from potentially alarming displacements (e.g. geo-hazard induced). MT-InSAR is also unable to measure rapid displacements exceeding phase unwrapping thresholds or movements occurring primarily in the north-south direction. Therefore, even dense PS coverage does not guarantee detection of all signs of structural degradation. Although the monitoring factor represents a simplification, it enables wide-scale estimation of routine remote monitoring benefits, and future studies could expand this factor to account for various SHM sensor types and MT-InSAR limitations.

The PS predictions provide a reasonable initial estimate of how well a bridge could be monitored, but they are not entirely accurate. Moreover, the predictions only indicate the expected number of PSs in a roughly $100 \times 100$ m pixel, and given that the availability of points is not uniform, optimal sensor placement is not guaranteed. Thus, a point on a structure that is actually affected by displacement might be missing from the structure. Additionally, the model used for PS prediction was trained only on descending data. Therefore, an underestimation might sometimes occur when the bridge is located such that it would be less visible for descending satellites but available for monitoring from ascending. Furthermore, the model for PS predictions was trained with data obtained through standardised MT-InSAR processing. Thus, a low PS count prediction does not necessarily

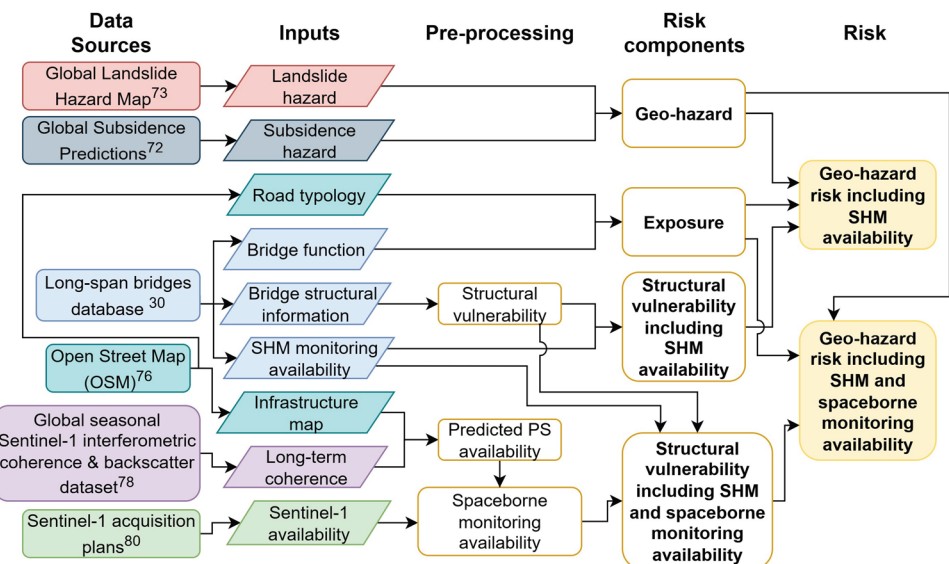

**Fig. 8 | Method flowchart for geo-hazard risk assessment with monitoring integration.** The methodology integrates data from multiple global sources to assess geo-hazard risk through three main components: hazard, exposure, and structural vulnerability. Hazard assessment uses landslide and subsidence datasets, whilst exposure evaluation is based on bridge functional characteristics. Structural vulnerability first considers bridge structural characteristics, then produces two assessments by integrating monitoring system availability: one incorporating only SHM sensor data from the bridge database, and another that additionally includes spaceborne monitoring potential estimated based on Sentinel-1 coverage and predicted measurement point density. The geometric mean of these three components produces overall bridge risk metrics, enabling comparison between scenarios with and without monitoring integration.

indicate that monitoring the bridge is impossible, but rather that it might require a more sophisticated approach. Conversely, bridges with high PS count predictions should be good targets for monitoring with standardised products such as the European Ground Motion Service (EGMS)[46]. This could reduce the monitoring cost as the data are already available and processed. Still, the EGMS uses water masks that remove some bridges, so even for European bridges, there might be a need for processing targeted for specific bridges. For bridges classified as unlikely to be monitored from space, alternative methods, such as those that include Distributed Scatterers or different SAR sensors, such as X-band satellites with higher spatial resolution, could be considered.

X-band is usually seen as superior for infrastructure monitoring due to its higher spatial resolution, providing PSs more densely, and because the lower spatial resolution of Sentinel-1 might smooth the spatial deformation pattern over the bridge[71]. However, previous studies have shown that whilst Sentinel-1's C-band sensor provides fewer PSs than the X-band sensor, the longer wavelength of the C-band is less susceptible to vibrations, resulting in more uniformly distributed PSs over bridges, including in regions like span centres not typically covered by X-band[18,64]. This feature, together with the free availability of C-band Sentinel data, makes it highly suitable for worldwide bridge monitoring. Nevertheless, future research could extend the risk assessment by incorporating the availability of X-band data or upcoming data sources such as L-band from the NISAR mission.

The proposed approach, despite its limitations, enables a global assessment of risk levels for long-span bridges and highlights the worldwide potential of MT-InSAR-based structural monitoring. By integrating monitoring availability into a comprehensive geo-hazard risk assessment framework, this study addresses a critical gap in previous research, which often excluded monitoring capabilities. The approach can support decision-making by identifying structures where risk mitigation measures are needed, where spaceborne monitoring is feasible, and which bridges should be prioritised for in-person inspections or sensor installations. By facilitating more efficient resource allocation based on specific risk profiles it allows stakeholders to focus on structures posing the greatest risk to communities,

contributing to safer and more resilient infrastructure systems. Therefore, the methodology can contribute to SDG 9 by improving infrastructure resilience and promoting equitable access to new technologies, research, and information, particularly benefiting developing countries.

## Methods

This study developed a methodology for assessing the geo-hazard risk of infrastructure. The key steps involved in the processing are outlined in Fig. 8. The method was applied to a catalogue of long-span bridges[30], utilising data on the functional characteristics of these bridges to assess their exposure. Subsidence and landslide hazards were evaluated to derive a geo-hazard value for each bridge. The vulnerability calculations were performed in two stages. First, the structural vulnerability of each bridge was assessed based on its structural characteristics, such as condition, material, typology, and age. Second, the availability of monitoring systems was integrated into the vulnerability metric. Spaceborne monitoring availability was determined using predictions of measurement point density and information on the coverage of Sentinel-1 data, while the presence of SHM sensors was known from the database of bridges. Including the knowledge about potential monitoring systems resulted in two structural vulnerability factors for each bridge—one considering only the currently installed in-situ sensors and another incorporating the potential for spaceborne monitoring. Finally, the geo-hazard, exposure, and structural vulnerability components were combined to produce an overall bridge risk metric.

### Long-span bridges database

The method was applied to a database of long-span bridges compiled in ref. 30. This database aggregates information from various online sources, providing comprehensive data on 751 bridges with main spans exceeding 150 m, as depicted in Supplementary Fig. 2. For the processing in this study, we removed duplicate bridges, resulting in 744 structures, and manually corrected the locations of some of them. Multiple parameters are recorded for each bridge, including construction material, bridge type, health status, and construction period, among others. Notably, the database also includes information on

reported structural health monitoring systems, which is particularly relevant to this study. However, it does not specify the particular SHM system installed on a structure. The majority of bridges in the database are located in North America, followed by Europe and Asia, with a significant proportion of the Asian bridges situated in China. The authors of the database noted that some regions might be under-represented due to the data collection being conducted using web searches and automatic language translators that could miss some entries from regions where English is not the primary language[30]. However, while the dataset may have inherent spatial biases, it remains the most comprehensive source of detailed global bridge information currently available, making it an invaluable resource for the worldwide bridge analysis in this article.

## Hazards

This article focuses on gradual ground deformation processes, specifically subsidence and slow-moving landslides, that can lead to bridge displacements detectable using MT-InSAR. These hazards typically evolve over time at rates compatible with the temporal sampling frequency of radar acquisitions, allowing for effective monitoring. In contrast, rapid landslides, which involve movement rates that exceed the revisit intervals of radar satellites, fall outside the scope of this study due to their incompatibility with MT-InSAR detection. While this work does not address such rapid events, future research could consider their inclusion in broader risk assessments.

Information on the expected magnitude of global subsidence in 2040 was derived from the predictions of ref. 72, which used subsidence susceptibility and groundwater depletion probability data to estimate the subsidence hazard level at a spatial resolution level of $1 \, km^2$. This hazard is classified on a six-level scale, describing the probability of subsidence. Water bodies are assigned a 'no data' value. The dataset is intended to highlight areas with high subsidence potential where further analysis might be necessary. Therefore, we used the data on the subsidence threat in 2040 to identify structures at risk of subsidence that would benefit from continuous monitoring systems. To facilitate a comparison between hazards included in this study, the subsidence hazard levels were normalised to a scale between 0 and 1, as shown in Supplementary Table 1. The maximum subsidence hazard value over each structure was used for analysis.

To assess the threat of landslides on structures, we utilised the 'Global Landslide Hazard Map' by ref. 73. This dataset combines landslide susceptibility, defined by terrain characteristics influencing the likelihood of failure, with data on the probability of events capable of triggering a landslide, such as rainfall and earthquakes. The map quantifies landslide hazards by estimating the average number of significant landslides per square kilometre and categorises the combined hazard from rainfall and earthquake triggers into four qualitative levels. For this study, landslide hazard levels were normalised (Supplementary Table 2), and the highest value associated with each structure was used in the analysis.

After obtaining the maximum subsidence and landslide normalised hazard levels, they were combined into a single geo-hazard value per structure using a geometric mean calculation with inversion and re-scaling procedures, as recommended by the INFORM framework[74], to ensure higher values correspond to greater hazard severity. The detailed methodology for this geometric mean calculation is described further in this manuscript in the risk calculation subsection.

## Exposure

Due to the absence of a standardised methodology applicable to the available bridge data, this study employs the bridge function as a measure of exposure. This approach aligns with the method proposed in ref. 75, which assigns exposure levels based on the type of body crossed by the bridge, considering the potential social and public consequences of bridge disruptions. However, since this information is not consistently available worldwide, we instead employed bridge functionality as a proxy for exposure, similar to the approach suggested by ref. 18. The long-span bridges database used in this study provides binary information on whether a bridge serves as a road, railway, footway, or a combination thereof. To further enhance the data from the bridge database, specific road types on bridges (i.e. highway, trunk, primary, and secondary roads) were extracted from the Open Street Map (OSM), an open-source geospatial database[76]. In cases where multiple road types were identified on a single bridge, the type corresponding to the highest exposure class was selected. The exposure levels were defined as outlined in Supplementary Table 3.

## Structural vulnerability

No universally accepted methodology currently exists for assessing the structural vulnerability of bridges. However, given that this study aimed to identify bridges with structural deficiencies in the risk assessment, we adopted an approach from the Italian Ministry of Infrastructure and Transport guidelines[75,77]. The vulnerability in these guidelines is determined using factors such as the level of degradation, construction period, design code class, and the structure's properties. This information was either available in the database used for the analysis or could be reasonably inferred. Supplementary Fig. 3 illustrates the logical flowchart used to assign vulnerability classes.

The bridge database used in this study categorises bridge condition into one of three states: 'good', indicating a structure not requiring immediate intervention; 'deficient', denoting a bridge in need of urgent structural repairs; and 'obsolete', where the asset is structurally sound but not currently utilised as originally intended. According to the Italian guidelines[75], degradation can be classified into five levels. For this analysis, structurally deficient bridges were assigned the highest degradation level. The database identified several issues for the remaining bridges, including moisture, wearing, corrosion, erosion, and cracking. Bridges with no additional issues were assigned the lowest degradation level; those with a single problem were rated as medium-low degradation, while bridges with multiple issues were rated at the medium level. Although the Italian guidelines define five degradation levels, the limited structural health information available in the bridge database allowed for the assignment of only four levels, with the medium-high degradation category remaining unused in this analysis. The most recent date between the original construction completion and the reconstruction year was used to determine the construction period. Since the design class was not provided in the database, the worst-case scenario was assumed per the guidelines, categorising all bridges as designed for civil loads only, corresponding to category II. Furthermore, since the design year was also unknown, the year construction commenced was used to assign classes: Class A for bridges whose construction began before 1990, Class B for those that started between 1990 and 2007, and Class C for bridges built after that.

Finally, the static scheme, materials, and span length were considered. The long-span bridge types in the database (arch, cable-stayed, cantilever, suspension, and truss) do not directly correspond to the static schemes explicitly defined in the Italian Guidelines. To overcome this constraint, the guidelines' recommendation to apply conservative vulnerability assessments when specific classifications are unavailable was followed[75]. For bridge types not explicitly covered in the guidelines, the highest vulnerability class was assigned as a conservative approach, with material-based adjustments following the guidelines' established hierarchy. These assumptions are summarised in Supplementary Table 4.

Following this process, each bridge was classified into one of five structural vulnerability classes: low, medium-low, medium, medium-high, and high, with corresponding numerical values $V$ of 0.2, 0.4, 0.6, 0.8, and 1.0, respectively.

## Monitoring capabilities

This paper introduces a method for including monitoring capabilities in structural vulnerability to enhance risk assessment and aid in prioritising resources for bridge inspections. Two types of monitoring are considered: SHM sensors and spaceborne monitoring. The monitoring capabilities are assessed to assign a monitoring factor, denoted as $f_{monitoring}$, which is then multiplied by the structural vulnerability $V$ to provide the integrated vulnerability value $V_{monitoring}$, as described in the following equation:

$$V_{monitoring} = \begin{cases} V \times f_{monitoring}, & \text{if } V \times f_{monitoring} < 1 \\ 1, & \text{otherwise} \end{cases} \quad (1)$$

The monitoring factor ranges from 1 (representing perfect monitoring) to 1.35 (indicating no monitoring). These values were inspired by the confidence factor in Eurocode-8[65] and the knowledge level in FEMA 356[66], both of which quantify confidence in the knowledge of building component properties and account for uncertainties in data collection. The availability of monitoring systems reduces the integrated structural vulnerability by providing higher confidence in time-variant structural assessment. In this way, bridges without monitoring are penalised with increased vulnerability, flagging them for priority consideration in risk-based monitoring strategies. Notably, if monitoring reveals deteriorating conditions, the physical structural vulnerability can be updated accordingly, leading to a more accurate overall risk evaluation.

Binary information regarding the availability of SHM sensors on each structure was acquired from the database used for the study. For this analysis, it was assumed that bridges equipped with an SHM system have perfect monitoring capabilities. Therefore, such bridges were assigned a monitoring factor of 1, while those without SHM sensors were assigned a factor of 1.35.

A more detailed methodology was required to evaluate the spaceborne monitoring capabilities. The steps of this process are summarised in Supplementary Fig. 4. The database of bridges used for the analysis provides only point-like coordinates for each structure. However, knowledge of the location of the whole length of the bridge was required for this study. To address this, roads and pathways along each bridge were identified using OSM data[76] and converted into polygons. A centre line was then created for each polygon and subsequently divided into five equal-length sections: the central span, two intermediate sections, and two end sections, to allow for a segment-specific analysis.

Next, the number of PSs per pixel was estimated using the method proposed in ref. 47. This involved downloading long-term coherence data from ref. 78 for summer (Northern Hemisphere) or winter (Southern Hemisphere), which are provided at a spatial resolution of approximately $100 \times 100$ m. Infrastructure maps were then generated using OSM data[76]. These coherence and infrastructure datasets were used as inputs to a pre-trained model to infer pixel-wise PS density for each pixel. This approach estimated the number of PSs for each roughly $100 \times 100$-m pixel that intersected with the bridge geometry, without applying additional buffer zones around the bridge structure.

To enable comprehensive spatial analysis of monitoring availability across bridge structures, each bridge was divided longitudinally along its centreline into five equal-length segments: two edge sections, two intermediate sections, and one central span. Two key statistics were calculated to evaluate PS availability for each bridge segment: (1) the proportion of pixels with at least one PS and (2) the number of PSs per 100 m. The proportion of pixels with at least one PS was used to assess the spatial distribution of PSs across a bridge, as uniform coverage is critical for reliable deformation analysis[18]. This statistic was calculated as the ratio of pixels containing at least one PS to the total number of pixels in a given segment. It ranged from 0 (indicating no PS coverage) to 1 (indicating uniform spatial distribution). The length of

the bridges in the database used for the analysis varies significantly. To ensure a fair comparison, the predicted number of PSs per segment was normalised by the length of the segment, providing the average number of PSs per 100 m. Low PS availability over a bridge limits the ability to perform a comprehensive deformation analysis[36]. Consequently, the higher the number of PSs per 100 m, the better the monitoring capabilities. These two statistics were used to derive the PS availability class for each segment, according to Table 1.

Finally, the weighted mean of the PS availability values across all segments was calculated, placing the highest emphasis on the central span:

$$A = 0.1 \times (A_{e1} + A_{e2}) + 0.25 \times (A_{i1} + A_{i2}) + 0.3 \times A_c \quad (2)$$

where $A$ denotes the overall PS availability value, $A_{e1}$ and $A_{e2}$ indicate the edge sections, $A_{i1}$ and $A_{i2}$ represent the intermediate sections, and $A_c$ corresponds to the central span. This adjustment accounted for the higher likelihood that PSs in the end sections may belong to nearby structures, such as buildings, which could skew the statistics[79].

After estimating the PS availability class for each bridge, the accessibility of Sentinel-1 data, based on the acquisition plans[80], was factored in to account for the SAR data coverage for each specific bridge. Sentinel-1 offers two primary repeat intervals: 6 and 12 days. Higher temporal sampling enables more detailed displacement analysis. Additionally, in some regions, Sentinel-1 provides imagery from ascending and descending geometries, while only one acquisition geometry is available in other areas. To comprehensively understand structural movement, data from both flight directions are preferred, as it allows displacement measurements to be decomposed into vertical and east-west components, providing more detailed insights[34,71]. Therefore, the highest monitoring capability is achieved when data from both flight directions are acquired every 6 days. In such cases, the spaceborne monitoring capabilities were assumed to be equal to the PS availability class, serving as a baseline. If Sentinel-1 obtains data from both flight directions but only every 12 days, the value was reduced by 0.1. If data acquisitions are made from only one flight direction, the monitoring capability was reduced by 0.2 if the interval between acquisitions is 6 days and by 0.3 if the temporal sampling is 12 days. For the plots, spaceborne monitoring was categorised into five levels: no monitoring [0, 0.2], low (0.2, 0.4], medium (0.4, 0.6], high (0.6, 0.8], and very high (0.8, 1.0].

Once the PS availability was integrated with temporal sampling and flight path availability of Sentinel-1 data into a spaceborne monitoring value for each bridge, it was re-scaled from the range [0,1] to [1.35,1] to create the monitoring factor. Scaling was performed to ensure that bridges that could not be monitored from space were assigned a factor of 1.35, while those with perfect spaceborne monitoring value received a factor of 1. To integrate SHM and spaceborne monitoring into one metric, a combined value was obtained by taking the lower of the two re-scaled monitoring factors.

## Geo-hazard risk

The risk was calculated by taking the geometric mean of hazard, exposure, and vulnerability, inspired by the methodology from the INFORM framework[74]. Since the geometric mean is typically smaller than the arithmetic mean, and to ensure that higher values—indicating a worse status for the element at risk—are appropriately rewarded, the INFORM framework suggests first inverting the values, calculating the geometric mean, and then reversing the values again. Additionally, values must be re-scaled to avoid computational instabilities, ensuring that all values remain positive. The steps involved in calculating the risk are as follows:

1.  Reverse the values of $x$, where $x$ represents hazard, exposure, or vulnerability, so that higher values indicate lower levels of the

corresponding risk component:

$$x_{reversed} = 1 - x \qquad (3)$$

2. Re-scale the values from the range [0,1] to [1,10] to prevent division by zero

$$x_{rescaled} = 9 \times x_{reversed} + 1 \qquad (4)$$

3. Calculate the geometric mean of hazard, exposure, and vulnerability, using the reversed and re-scaled values:

$$mean = \sqrt[n]{x_{rescaled_1} \times x_{rescaled_2} \times \cdots \times x_{rescaled_n}} \qquad (5)$$

4. Re-scale the mean back to the range [0,1]:

$$mean_{rescaled} = \frac{1}{9}(mean - 1) \qquad (6)$$

5. Invert the value again so that higher values reflect greater risk:

$$mean_{final} = 1 - mean_{rescaled} \qquad (7)$$

The same methodology was applied to calculate geo-hazard, using the geometric mean of subsidence and landslide hazards.

## Data availability

The datasets used in this study are publicly available from the following sources: long-term coherence data were obtained from the Sentinel-1 Global Coherence and Backscatter dataset available at https://registry.opendata.aws/ebd-sentinel-1-global-coherence-backscatter/; Sentinel-1 data availability was derived from acquisition plans available at https://sentinels.copernicus.eu/web/sentinel/copernicus/sentinel-1/acquisition-plans; Global Landslide Hazard Map is available at https://datacatalog.worldbank.org/dataset/global-landslide-hazard-map; and Potential Global Subsidence data for 2040 are available at https://figshare.com/articles/dataset/Global_Subsidence_Maps/13312070/1. The long-span bridges database is available upon request from the authors of ref. 30 (https://doi.org/10.1080/15732479.2019.1639773). Map data are copyrighted by OpenStreetMap contributors and available from https://www.openstreetmap.org. Pre-processed and normalised versions of the landslide and subsidence datasets, processed Sentinel-1 availability shapefiles, regional Persistent Scatterer predictions, risk assessment results, and source data for all figures are archived in the Zenodo repository associated with this paper (https://doi.org/10.5281/zenodo.15797029). Source data are provided with this paper.

## Code availability

The code used in this study is available at https://github.com/dominika-malinowska/bridge-risk-assessment-insar and archived on Zenodo (https://doi.org/10.5281/zenodo.15814218).

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

## Acknowledgements

We thank Colin Caprani for providing the long-span bridge database used in this study. This publication is part of the Vidi project InStruct, project number 18912, financed by the Dutch Research Council (NWO) (GG). Part of this work was performed at the University of Houston under a contract with the Commercial Smallsat Data Scientific Analysis Program of NASA (NNH22ZDA001N-CSDSA) and the NASA Decadal Survey Incubation Program: Science and Technology (NNH21ZDA001N-DSI) (PM).

## Author contributions

D.M.: Conceptualisation, data curation, formal analysis, investigation, methodology, validation, visualization, writing—original draft, writing—review and editing; P.M.: Conceptualisation, funding acquisition, methodology, writing—review and editing; C.R.: Conceptualisation, methodology, writing—review and editing; C.B.: Conceptualisation, methodology, writing—review and editing; G.G.: Conceptualisation, funding acquisition, methodology, resources, supervision, writing—review and editing.

## Competing interests

The authors declare no competing interests.
