## [Transparent Peer Review file · Nature Communications]

Global geo-hazard risk assessment of long-span bridges enhanced with InSAR availability

Corresponding Author: Ms Dominika Malinowska

Version 0:

Reviewer comments:

Reviewer #1

(Remarks to the Author)

The manuscript concerns a methodology for risk assessment for bridges accounting for potential MTInSAR monitoring. The authors claim they present a methodology for qualitative risk assessment suitable for geohazards, accounting for the availability of a satellite-based or sensor-based monitoring which impact the vulnerability component. In this reviewer's opinion, the novelty of the manuscript is related to the results (Figures 1 to 6) discussing the potential suitability of a MTInSAR-based monitoring for long-span bridges all over the world. The methodology for risk assessment, although potentially efficient for large-scale monitoring, is basic and retraces existing class-based approaches such as the one proposed by the "Italian Guidelines for existing bridges" (mentioned by the authors).

In this reviewer's opinion, the manuscript requires an extensive revision considering the following comments.

- In this reviewer's opinion, the methodology can be hardly defined as generically "multi-hazard". Indeed, only geohazards, specifically landslide and subsidence, are considered. The term "multi-hazard" in the title may be misleading for readers, as it suggests that the manuscript presents a risk assessment methodology capable of addressing multiple hazard sources, such as hydraulic, seismic, and structural factors.
- The term of "long-span" bridges is not clear in the beginning of the manuscript (introduction and results). The authors should clarify in the introduction which bridge types are included and the considered span length limits in the adopted database. Also the database should be presented, reporting statistics (e.g. on the bridge typologies included, materials, presence of sensors) which can be useful for reading the results in figures 1-7.
- Since the methodology adopted MTInSAR ad concerns monitoring bridges to geo-hazards, the state-of-the-art should be modified and extended including specific references on the issues related the structural response of bridges under landslides/subsidence/settlements (see, for example, <https://doi.org/10.1007/s10518-022-01544-3>, <https://doi.org/10.1016/j.soildyn.2025.109335>, <https://doi.org/10.1016/j.soildyn.2024.109079> and others).
- In the risk assessment methodology, the authors should comment on the differences of the hazard module provided in the study and the one proposed for geo-hazards by the Italian Guidelines on the structural safety of existing bridges, which is used by the authors for the definition of the vulnerability module. Also, the vulnerability model adopted in this study "Supplementary figure 3" does not include the last parameter "magnitude of the interference" which is specifically proposed by the Guidelines for the vulnerability to geo-hazard. Why?
- Supplementary Table 4. It is not clear the similitude in structural vulnerability between 1) "simply supported girders" in the Italian Guidelines and "cable-stayed"; 2) "truss" in the Italian Guidelines "continuous girder"; 3) "Gerber bridges" in the Italian Guidelines with "suspension bridges". This reviewer strongly suggests a revision of this table and recommends an appropriate definition of the typological vulnerability associated with different static schemes (rather than using the classes reported in the Italian Guidelines). Note also that the vulnerability of a static scheme of a given material, according to the Italian Guidelines" is defined considering the material of the superstructure. Therefore, the last part of the section "structural vulnerability" should be revised.
- L131 "The monitoring factor ranges from 1 (representing perfect monitoring) to 1.35 (indicating no monitoring)". The authors justify the coefficient 1.35 referring to the confidence factor related to a given knowledge level to be assigned to mechanical parameter in the mentioned codes. The connection of the concepts for what this coefficient is conceived (a safety coefficient related to the lack of knowledge) and the scope for which it is used in this study (increasing the vulnerability related to the absence of monitoring strategy) is not clear and should be explained.
- In the vulnerability module, the monitoring parameter considers the presence of generic sensors which are not specifically defined in the manuscript. Please specify which sensor types are considered as potential factors for decreasing the bridge vulnerability. Note that there exist sensors collecting data that could be not related to monitoring geohazards, and, therefore,

useless for the scope of the methodology.

- Please clarify the term V in Equation 1. Is it the vulnerability class? This is not clear.

- L222 - please provide further information on the way to define the length of the five segments in which the bridge longitudinal axis is divided.

- L367 "Correlations between bridge PS availability and bridge characteristics were analysed. Whilst the physical properties of materials can impact the strength of the SAR signal return, no clear correlation between the bridge material and its potential for monitoring was found." In this reviewer's opinion, the bridge material (concrete or steel) could be not influencing for the suitability for spaceborne monitoring since generally satellites observe the roadway surface on the bridge (generally asphalt?), rather than the material of the superstructure or the substructure. Is it correct? Provide clarifications.

- L378 "This could be because suspension bridges are more susceptible to vibrations, especially in the central span, potentially causing a loss of coherence between acquisitions." This represent a limitation of MTInSAR since generally the central part of the bridge (high span length in the central part) is more critical than the approaching bridge sections.

Reviewer #2

(Remarks to the Author)

Dear Authors,

Based on my review of the paper "Global multi-hazard risk assessment of long-span bridges enhanced with remote sensing availability," this paper proposes integrating radar remote sensing technology for the risk assessment of long-span bridges on a global scale. The findings could significantly contribute to the risk management of bridges worldwide and align with the SDGs for urban and infrastructure sustainable development. The paper is well-designed and organized. However, the current version is not suitable for publication. Therefore, I would like to provide some comments on technical and experimental aspects. I hope these comments can help improve the quality of this manuscript.

In total, there are 20 comments.

20. Overall, the authors claim to have proposed a novel method for assessing bridge risk on a global scale. However, as I understand it, the authors have integrated potentially usable remote sensing technology, such as InSAR, into an existing assessment framework for bridge risk assessment, where the ability to be monitored plays an important role. It is well-known that InSAR is a globally usable technology, though it may be limited by certain issues. The authors analyze comparison results before and after incorporating InSAR measurements for global bridge risk assessment, noting that risk accuracy/reliability improves with this integration. However, after reviewing this manuscript multiple times, it remains unclear what specific types of risks the authors analyze and how, as they only highlight "multi-hazard" throughout the title, content, and results. Although the landslide, subsidence, and structure deformation are highlighted. Additionally, it is unclear whether bridge deformation is measurable, even with many PS measurements on the bridge, because bridge deformation differs significantly from other hazards. Even if deformation is captured, it may not be of concern to engineers, such as thermal dilation or vibration. I hope the authors can redesign the paper to eliminate these ambiguities.

My other comments, as shown below,

1. According to my reading, the manuscript could benefit from a clearer logical flow. Some parts that belong in the discussion section appear in the results section, and the transitions between sections need better connectivity.

2. After I finished browsing through the entire paper, I noted the length of the bridge investigated is beyond 150 m. In my opinion, it would be better to indicate the specific length of the bridge explored at the beginning of this paper. So that the readers can know more information without question in advance.

3. The title may not accurately reflect the content, which is overdefined, as it does not fully address multi-hazard risks and focuses solely on InSAR measurement availability rather than all remote sensing technologies.

3. In lines 48 to 51, it would be better to consider revising the statement following the introduction of vulnerability for better clarity and flow.

4. While Sentinel SAR technology offers regular monitoring, it does not provide real-time data. Avoid overstating its capability for near-real-time monitoring, especially in structural health monitoring contexts.

5. In Line-97 to 102, I don't understand the causal relationship in this paragraph. Is the author trying to emphasize their previous work on predicting PS points in the reference of 47?

6. In Line-122 to 126, the claim that remote sensing has not been used to reduce in-person surveys in bridge risk assessment is inaccurate. Many studies have explored bridge deformation using InSAR. Additionally, this study does not fully address this issue. Could you please identify the real gap for this study?

7. In line 141, the authors claim that they already evaluate the subsidence and landslide hazards using global datasets. The term "evaluates" may not be appropriate, as the study integrates multi-hazard results from open sources and InSAR availability rather than conducting original evaluations.

8. In line 175 to 178, there is a discrepancy between the number of indices mentioned in the text (three) and those in Table 1 (two). Please clarify.
9. The placement of the section on factors affecting PS availability seems awkward. Consider moving it to the discussion section.
10. In line-210, the claim that there is no relationship between bridge material and PS availability needs further discussion or supporting references. Because it directly affects your findings and conclusions.
11. In line-229 to 238, please provide more explanation on why bridges-oriented E-W have higher PS availability, as this conflicts with common understanding, especially for certain bridge types. Also, clarify why azimuth does not significantly affect PS availability. So, could you please provide some detailed information?
12. In Figure 1, please provide more context for terms like "s/c" in Figures 1 and 7a to avoid confusion for readers from other disciplines.
13. For the results in Figure 3a, Figure 4a, Figure 5a and Figure 6, in my opinion, it is improper to use a small number of the bridge from the region like Africa and Oceania as the representative. It may lead to biased conclusions; especially the reduction percentage of risk level may have some error and be biased. Even you have achieved the bridge information from the open-source data, could you please spend more time collecting more information about them? The corresponding experiment should be updated or redesigned.
14. The section title of "including monitoring in the risk assessment" clarifies what type of monitoring is included in the risk assessment section.
15. In line 338 to 350, consider moving the discussion on PS availability to the discussion section where it is addressed, which has been stated in the previous questions.
16. In line 394, could you please explain why vertical deformation is more critical for bridge structural health? Also, I have checked some papers indicating that the bridge deformation can reach up to 1 meter as well as the lateral displacement. On the other hand, for such a kind of deformation, can the InSAR truly capture it? In my opinion, like we highlighted before, the author should be careful to introduce or clarify what kind of bridge deformation you are investigating.
17. In the section on hazards, describe how you integrated hazard information for different types, like landslides and subsidence, given their varying spatial resolutions.
18. Include a new section detailing the open-source datasets used in your study.
19. In line 762, define "per pixel" and clarify the type of pixel. Specify if you analyzed PS availability over a buffer zone centered on the bridge.

Reviewer #3

(Remarks to the Author)

The manuscript presents a novel, globally scalable methodology for multi-hazard risk assessment of long-span bridges, integrating remote sensing (MT-InSAR) availability into structural vulnerability estimation. The work is timely and relevant, offering valuable insights into how satellite-based monitoring can supplement traditional Structural Health Monitoring (SHM) systems, particularly in underserved regions. The manuscript is generally well written and scientifically sound. However, I believe that several critical methodological aspects require clarification or further development before the work can be considered for publication.

Major Comments

1. Unclear Role of InSAR in Structural Health Monitoring

The manuscript does not clearly articulate how spaceborne MT-InSAR can reliably inform on the structural health of bridges, particularly given the fragile failure mechanisms often associated with such structures. It remains ambiguous whether the authors aim to detect long-term displacements due to external factors (e.g., subsidence, landslide) or direct signs of structural deterioration. A more explicit discussion is required on what type of bridge behavior is realistically observable via MT-InSAR, considering its spatial and temporal resolution, and what kind of deterioration mechanisms it can meaningfully capture.

2. Limited Hazard Scope and Oversimplified Hazard Integration

Only two hazards—subsidence and landslides—are considered, while many regions are predominantly affected by acute hazards such as earthquakes, floods, or hurricanes. This limited hazard spectrum severely restricts the generalizability of the approach. Furthermore, the manuscript treats these two hazards as independent and additive, ignoring potential interdependencies or cascading effects (e.g., subsidence-induced slope failure). This is a substantial limitation in a multi-hazard context and should be more critically discussed. The methodology should be extended or at least conceptually prepared to handle compound risks.

3. Ambiguity in Hazard Calculation

The description of how landslide and subsidence hazard levels are incorporated into the risk metric is unclear in the Methods section. It is essential to clarify how these datasets are normalized, weighted, or combined in the final risk computation. Figure 8, intended to summarize the methodology, is overly complex and should be simplified or supplemented with clearer narrative explanations.

4. Arbitrary Monitoring Reduction Factors

The methodology applies vulnerability reduction factors (1.35 to 1.0) based on SHM or InSAR monitoring availability. However, the rationale behind these numerical values is not empirically justified. This approach assumes that the mere presence of monitoring reduces risk, without considering monitoring quality, continuity, or resolution. The authors should explain how these factors were determined and whether any sensitivity analysis was conducted to assess their impact on the results.

5. Use of Static and Coarse-Resolution Input Data

Exposure and vulnerability estimations rely on static, low-resolution, or indirectly inferred data (e.g., bridge typologies, OSM-derived road functions, coarse age brackets). The implications of this data coarseness on the reliability of the risk results are not discussed. The authors should elaborate on how these limitations affect the statistical robustness and regional applicability of their method.

6. Overstated Effectiveness of MT-InSAR Monitoring

The paper emphasizes the wide potential of MT-InSAR but does not adequately address its limitations, including:

- o Susceptibility to false positives/negatives due to decorrelation and atmospheric effects;
- o Inability to detect abrupt or high-frequency structural failures (e.g., due to earthquakes);
- o Oversimplified assumption that PS density correlates directly with effective monitoring, without accounting for structural geometry, orientation (e.g., N-S aligned bridges), or radar-specific limitations.

These aspects should be discussed more critically, particularly in the context of decision-making based on such data.

7. Uncertainty Quantification

While monitoring is framed as reducing “epistemic uncertainty,” the manuscript does not explain how uncertainty is quantitatively treated or propagated through the risk model. This omission undermines the credibility of the final risk scores and limits their usefulness in prioritization or policy decisions. The authors should at least discuss the current limitations in this regard and propose potential directions for future incorporation of uncertainty modeling.

Minor Comments

- Lines 617–618: The manuscript describes landslides as gradual processes. This is misleading, as many landslides—especially those triggered by rainfall or seismic events—can be very rapid. Please revise for accuracy.
- Line 689: The text states that degradation is classified into five levels, yet Figure S3 seems to depict only four. Please clarify or correct the inconsistency.
- Line 670 and Figure S3: The description of structural vulnerability should align more closely with the logic and classification levels presented in Figure S3. Greater consistency would improve clarity.

Version 1:

Reviewer comments:

Reviewer #1

(Remarks to the Author)

This reviewer maintains some reservations regarding the feasibility and reliability of the approach related to the arbitrary selection of the monitoring factors (see comment R1.7) and the type of sensor adopted to account for the presence of in-contact monitoring (comment R1.8). However, these issues may be addressed in further research developments. In this reviewer's opinion, the authors' responses have addressed the main issues raised in the original version of the manuscript and it does not require further modifications.

(Remarks on code availability)

Reviewer #2

(Remarks to the Author)

Dear Authors,

Thank you for your efforts in addressing my comments. It is great to see that you have revised all the figures and included statistical information. The revised manuscript has improved significantly and is much clearer than before. However, there are still some additional points that need to be addressed to avoid confusion for future readers. My comments are listed below:

1. On page-line 67, it is not accurate to say that periodic inspections are insufficient, as the satellite observations used in this study are also a form of periodic inspection. Please clarify the 'real' distinction between traditional inspections and satellite-based monitoring.
2. On page-line 194, the phrase "reduce global geo-hazard risk" may overstate the impact of this study. In my opinion, this research primarily revises the assessment strategy to potentially improve risk level definitions but does not directly reduce

global geo-hazard risks. Please consider rephrasing this to more accurately reflect the contributions.

3. Regarding Figure 3a, could you explain why spaceborne monitoring in North America appears to be the least (49%) compared to other regions? Please provide the reason for this observation. Is that due to the Sentinel-1 coverage or not?

4. For Figure 7b, the current description in the text is somewhat difficult to understand. Could you please revise it to make it clearer or consider rephrasing it in a different way?

5. On page-line 555, I am unsure if it is appropriate to add a subtitle for the following section, such as "Limitations of this study." However, this is ultimately up to the authors to decide.

6. Regarding Supplementary Figure 3, the authors assigned a higher weight to bridges built after 1980 (which may increase the degradation speed level) and a lower weight to those built before 1945. Could you please provide justification or references to support this weighting scheme? It would be helpful for readers to understand the rationale behind this approach.

(Remarks on code availability)

I have reviewed the code, and it works well for me.

Reviewer #3

(Remarks to the Author)

The authors have satisfactorily addressed all major and minor comments raised in the previous review round. The revised manuscript now presents a clear, well-structured, and methodologically sound study, offering a novel and globally scalable framework for integrating MT-InSAR monitoring potential into geo-hazard risk assessment of long-span bridges.

In detail:

- The introduction and discussion now explicitly delineate which bridge behaviours can be observed with MT-InSAR, and the limitations related to resolution, displacement direction, and rapid events. This balances enthusiasm for the technique with a realistic appraisal of its applicability.
- The terminology has been refined from "multi-hazard" to "geo-hazard," with a clear justification for focusing on landslides and subsidence, and a well-reasoned explanation of why cascading effects are left for future work.
- The Methods section and Figure 8 have been revised for greater clarity, detailing the normalisation and combination of hazard datasets.
- The rationale for the 1.35 coefficient is now linked to established engineering codes, and a sensitivity analysis confirms its minimal effect on risk classification.
- The discussion now explicitly addresses the coarse resolution and static nature of input datasets, potential regional biases, and implications for interpretation.
- Terminology, figure alignment, and classification levels have been harmonised.

The manuscript reads well, is logically organised, and is accessible to a multidisciplinary audience. Figures are generally clear, although Figure 8, while improved, could still benefit from slightly larger fonts for legibility in print. I did not identify any remaining substantive methodological issues or inconsistencies. Accept in present form, subject only to minor editorial polishing by the journal.

(Remarks on code availability)

REVIEWER COMMENTS

ID	Reviewer #1 (Remarks to the Author):	Response
R1.1	The manuscript concerns a methodology for risk assessment for bridges accounting for potential MTInSAR monitoring. The authors claim they present a methodology for qualitative risk assessment suitable for geohazards, accounting for the availability of a satellite-based or sensor-based monitoring which impact the vulnerability component. In this reviewer's opinion, the novelty of the manuscript is related to the results (Figures 1 to 6) discussing the potential suitability of a MTInSAR-based monitoring for long-span bridges all over the world. The methodology for risk assessment, although potentially efficient for large-scale monitoring, is basic and retraces existing class-based approaches such as the one proposed by the "Italian Guidelines for existing bridges" (mentioned by the authors).	We appreciate the Reviewer's feedback and have addressed the detailed comments point by point below.

	In this reviewer's opinion, the manuscript requires an extensive revision considering the following comments.	
R1.2	- In this reviewer's opinion, the methodology can be hardly defined as generically "multi-hazard". Indeed, only geohazards, specifically landslide and subsidence, are considered. The term "multi-hazard" in the title may be misleading for readers, as it suggests that the manuscript presents a risk assessment methodology capable of addressing multiple hazard sources, such as hydraulic, seismic, and structural factors.	We thank the Reviewer for this observation. To accurately reflect the scope of our work, we have revised the manuscript to replace all references to "multi-hazard" with "geo-hazard," including in the title.
R1.3	- The term of "long-span" bridges is not clear in the beginning of the manuscript (introduction and results). The authors should clarify in the introduction which bridge types are included and the considered span length limits in the adopted database. Also the database should be presented, reporting statistics (e.g. on the bridge typologies included, materials, presence of sensors) which can be useful for reading the results in figures 1-7.	The introduction has been updated to include the definition of long-span bridge and now the sentence is as follows: "Applied to a global dataset of 744 long-span bridges, i.e. those with main spans exceeding 150 meters, our methodology employs open-source subsidence and landslide hazards global datasets, calculates exposure metrics based on bridge functional characteristics, and combines these with the enhanced vulnerability factors to generate comprehensive risk profiles that inform monitoring decisions." (Lines 199-206) In the Results section, the plots include statistics on the database, e.g. the distribution of bridges across different typologies and regions. References to the original study that compiled the database have been included for completeness.
R1.4	- Since the methodology adopted MTInSAR ad concerns monitoring bridges to geo-hazards, the state-of-the-art should be modified and extended including specific references on the issues related the structural response of bridges under landslides/subsidence/settlements (see, for example, https://doi.org/10.1007/s10518-022-01544-3, https://doi.org/10.1016/j.soildyn.2025.109335, https://doi.org/10.1016/j.soildyn.2024.109079 and others).	We appreciate the Reviewer's suggestion. In response, we have expanded the state-of-the-art section to include a dedicated paragraph discussing the structural response of bridges affected by landslides and subsidence. The revised text incorporates the suggested references. The new paragraph is as follows: "Monitoring is crucial in the context of landslides and ground subsidence, which present unique challenges for bridge management due to their progressive

nature and the complexity of their structural response. Landslides cause foundation and abutment sliding and undercutting, impact piers, and can even affect overall structural integrity^{58,59}. Ground subsidence induces differential settlements that cause bridge displacement in horizontal and vertical dimensions, leading to structural tilting and torsion, and causing visible damage, including abutment deformation, deck cracking, and span rotational twisting⁶⁰. The structural response involves complex stress redistribution where the superstructure must accommodate both subsidence-induced stresses and normal loads amongst misaligned supports, with time-dependent interactions between ongoing ground movement and bridge material behaviours exacerbating structural deterioration and potentially causing impact damage between bridge elements⁶¹. These geo-hazards can exhibit coupled effects with seismic activity, where pre-existing ground displacement or landslides amplify seismic response and increase overall structural vulnerability^{59,61,62}. The impact of localised displacement caused by geo-hazards on bridges' structural health, such as that affecting piers, can be more difficult to detect during periodic inspections compared to the effect of more global displacement that affects the whole structure and often results in more visible functionality issues⁶¹. Thus, remote sensing techniques are particularly valuable for geo-hazard-affected bridges, as they could detect localised deformation patterns that might be overlooked in routine inspections and enable frequent monitoring of progressive movements. (Lines 139-169)

58. Salciarini, D. et al. Landslide-bridge interaction: Insights from an extensive database of Italian case studies. *Int. J. Disaster Risk Reduct.* 114, 104983, DOI: 10.1016/j.ijdr.2024.104983 (2024).

59. Lian, Q., Chen, L., Dang, X., Zhuo, W. & Li, C. Dynamic response and fragility of mountain bridges under the coupled effects of transverse earthquakes and landslides. *Soil Dyn. Earthq. Eng.* 188, 109079, DOI: 1185 10.1016/j.soildyn.2024.109079 (2025).

60. Zhang, C., Wu, K., Huang, S., Li, L. & Rao, X. Study on the simulation of bridge deformation in a mining subsidence area. *Sci. Reports* 15, 529, DOI: 10.1038/s41598-024-84220-7 (2025). Publisher: Nature Publishing Group.

		61. Nettis, A., Di Mucci, V. M., Ruggieri, S. & Uva, G. Seismic fragility and risk assessment of isolated bridges subjected to pre-existing ground displacements. Soil Dyn. Earthq. Eng. 194, 109335, DOI: 10.1016/j.soildyn.2025.109335 (2025). 62. Mantakas, A., Tsatsis, A., Loli, M., Kourkoulis, R. & Gazetas, G. Seismic response of a motorway bridge founded in an active landslide: a case study. Bull. Earthq. Eng. 21, 605–632, DOI: 10.1007/s10518-022-01544-3 (2023). “
R1.5	- In the risk assessment methodology, the authors should comment on the differences of the hazard module provided in the study and the one proposed for geo-hazards by the Italian Guidelines on the structural safety of existing bridges, which is used by the authors for the definition of the vulnerability module. Also, the vulnerability model adopted in this study “Supplementary figure 3” does not include the last parameter “magnitude of the interference” which is specifically proposed by the Guidelines for the vulnerability to geo-hazard. Why?	In our study, the definition of hazard aligns with the general understanding of it as the potential source of damage to assets. While the Italian Guidelines on the structural safety of existing bridges provide a comprehensive framework for various geo-hazards, our analysis focuses specifically on landslides and subsidence at a national scale. Consequently, a direct one-to-one correspondence between our hazard module and the broader categorization used in the Guidelines is not appropriate, as the scope and type of hazards considered differ. Regarding the “magnitude of the interference” parameter referenced in the Guidelines, we agree that this is a relevant factor in detailed, local-scale assessments where the spatial extent of hazard-structure interaction can be meaningfully differentiated. However, our analysis is conducted at a coarse spatial resolution where it is not feasible to resolve partial or progressive impacts on individual structures. Instead, we adopt a conservative assumption that any intersecting landslide or subsidence event affects the structure in its entirety. Under this assumption the “magnitude of the interference parameter” would not vary across our dataset and consequently was excluded from the vulnerability model, as including it would imply a level of spatial resolution not supported by the available data.
R1.6	- Supplementary Table 4. It is not clear the similitude in structural vulnerability between 1) “simply supported girders” in the Italian Guidelines and “cable-stayed” ; 2) “truss” in the Italian Guidelines “continuous girder” ; 3) “Gerber bridges” in the Italian Guidelines with “suspension bridges”. This reviewer strongly suggests a revision of this table and recommends an appropriate definition of the typological	We thank the reviewer for highlighting this issue. We acknowledge that there is no direct structural equivalence between the long-span bridge types in our database and the static schemes explicitly defined in the Italian Guidelines, as the Italian Guidelines do not provide specific vulnerability classifications for cable-stayed, suspension, and cantilever bridges. To address this gap, we have revised Supplementary Table 4 specifying the assumption used, also following the

	vulnerability associated with different static schemes (rather than using the classes reported in the Italian Guidelines). Note also that the vulnerability of a static scheme of a given material, according to the Italian Guidelines” is defined considering the material of the superstructure. Therefore, the last part of the section “structural vulnerability” should be revised.	Guidelines' own recommendation to apply conservative vulnerability assessments when specific classifications are unavailable. The “Structural vulnerability” section has been modified accordingly: “Finally, the static scheme, materials, and span length were considered. The long-span bridge types in the database (arch, cable-stayed, cantilever, suspension, and truss) do not directly correspond to the static schemes explicitly defined in the Italian Guidelines. To overcome this constraint, the guidelines' recommendation to apply conservative vulnerability assessments when specific classifications are unavailable was followed⁷⁵. For bridge types not explicitly covered in the guidelines, the highest vulnerability class was assigned as a conservative approach, with material-based adjustments following the guidelines' established hierarchy. These assumptions are summarised in Supplementary Table 4.” (Lines 886-905) Importantly, the proposed framework remains flexible to accommodate alternative methodologies to assess structural vulnerability, whenever available.
R1.7	- L131 “The monitoring factor ranges from 1 (representing perfect monitoring) to 1.35 (indicating no monitoring).”. The authors justify the coefficient 1.35 referring to the confidence factor related to a given knowledge level to be assigned to mechanical parameter in the mentioned codes. The connection of the concepts for what this coefficient is conceived (a safety coefficient related to the lack of knowledge) and the scope for which it is used in this study (increasing the vulnerability related to the absence of monitoring strategy) is not clear and should be explained.	We have extended the discussion section to further clarify the rationale behind the selected value of the coefficient. Additionally, we have added a sensitivity analysis to show how the variation of this value would impact the final risk score. This is covered in the following text added to the discussion: “The monitoring factor used in structural vulnerability calculations reflects the assumption that the availability of remotely sensed data allows more frequent structural health updates, thereby reducing uncertainty in the inherently time-variant structural vulnerability and consequently decreasing overall bridge risk. Monitoring improves knowledge about the structure, hence we adopted a coefficient value of 1.35, commonly used in building codes and standards, such as Eurocode-8⁶⁵ and FEMA 356⁶⁶, to account for uncertainties associated with (lack of) knowledge about building materials. Although this is a standardised value, it has not previously been applied in this context. Therefore, to fully understand the impact of this value selection on the final risk score, we performed a sensitivity

		analysis. Supplementary Figure 5 shows that changes due to different monitoring factor selections would be minor, with few bridges changing their assigned risk category." (Lines 563-591)
R1.8	- In the vulnerability module, the monitoring parameter considers the presence of generic sensors which are not specifically defined in the manuscript. Please specify which sensor types are considered as potential factors for decreasing the bridge vulnerability. Note that there exist sensors collecting data that could be not related to monitoring geohazards, and, therefore, useless for the scope of the methodology.	Unfortunately, the bridge database used in this study does not specify the type of sensor installed on the bridge, which indeed is a limitation of the method. We have updated the discussion to clearly state this constraint: "Moreover, the bridge database used in the study does not specify SHM sensor types, so our assumption of "ideal" monitoring may overestimate its capabilities. Additionally, MT-InSAR has inherent limitations requiring expert interpretation to distinguish expected movements (e.g., thermal dilation) from potentially alarming displacements (e.g., geo-hazard induced). MT-InSAR is also unable to measure rapid displacements exceeding phase unwrapping thresholds or movements occurring primarily in the north–south direction. Therefore, even dense PS coverage does not guarantee detection of all signs of structural degradation. Although the monitoring factor represents a simplification, it enables wide-scale estimation of routine remote monitoring benefits, and future studies could expand this factor to account for various SHM sensor types and MT-InSAR limitations." (Lines 639-654)
R1.9	- Please clarify the term V in Equation 1. Is it the vulnerability class? This is not clear.	Additional explanation has been added at the end of Structural Vulnerability methodology subsection, as follows: "Following this process, each bridge was classified into one of five structural vulnerability classes: Low, Medium-Low, Medium, Medium-High, and High, with corresponding numerical values V of 0.2, 0.4, 0.6, 0.8, and 1.0, respectively." (Lines 906-910)
R1.10	- L222 - please provide further information on the way to define the length of the five segments in which the bridge longitudinal axis is divided.	The methodology section has been extended to include an explanation of the division into the segments, as follows:

		"To enable comprehensive spatial analysis of monitoring availability across bridge structures, each bridge was divided longitudinally along its centreline into five equal-length segments: two edge sections, two intermediate sections, and one central span." (Lines 966-970)
R1.11	- L367 "Correlations between bridge PS availability and bridge characteristics were analysed. Whilst the physical properties of materials can impact the strength of the SAR signal return, no clear correlation between the bridge material and its potential for monitoring was found.". In this reviewer's opinion, the bridge material (concrete or steel) could be not influencing for the suitability for spaceborne monitoring since generally satellites observe the roadway surface on the bridge (generally asphalt?), rather than the material of the superstructure or the substructure. Is it correct? Provide clarifications.	We thank the reviewer for this thoughtful comment and acknowledge their concern. Indeed, considering only the correlation between the material of the bridge's main structure and PS availability might be an oversimplification. Often it is the superstructure or additional equipment installed on the bridge, such as signs or lampposts, that reflects the signal, as the asphalt usually acts like a smooth surface and reflects the backscatter echo away from the satellite (https://www.euspaceimaging.com/wp-content/uploads/backscatter_smooth_surface-1-768x509.png). We have expanded the discussion to include this explanation: "Correlations between bridge PS availability and bridge characteristics were analysed. Whilst the physical properties of materials can impact the strength of the SAR signal return, no clear correlation between the bridge material and its potential for monitoring was found. This could be attributed to the fact that bridges are typically constructed from steel and concrete, both of which are characterised by good reflective properties, or it might be because the radar signal is primarily reflected by the superstructure or additional equipment installed on the bridge, such as signs or lampposts." (Lines 439-441)
R1.12	- L378 "This could be because suspension bridges are more susceptible to vibrations, especially in the central span, potentially causing a loss of coherence between acquisitions.". This represent a limitation of MTInSAR since generally the central part of the bridge (high span length in the central part) is more critical than the approaching bridge sections.	We have updated the discussion to specify this limitation: "This is due to suspension bridges being more affected by vibrations, especially in the central span, that can cause loss of coherence and consequently decreased availability of PS⁶³. Although these bridges have fewer points overall, edge sections are monitored relatively well, suggesting that MT-InSAR could still provide meaningful information about the structural health of bridge piers and abutments. However, the limited availability of PSs in the central span represents a limitation of MT-InSAR usability for bridge monitoring." (Lines 446-456)

--	--	--

ID	Reviewer #2 (Remarks to the Author):	Response
R2.1	Dear Authors, Based on my review of the paper "Global multi-hazard risk assessment of long-span bridges enhanced with remote sensing availability," this paper proposes integrating radar remote sensing technology for the risk assessment of long-span bridges on a global scale. The findings could significantly contribute to the risk management of bridges worldwide and align with the SDGs for urban and infrastructure sustainable development. The paper is well-designed and organized. However, the current version is not suitable for publication. Therefore, I would like to provide some comments on technical and experimental aspects. I hope these comments can help improve the quality of this manuscript. In total, there are 20 comments.	We thank the Reviewer for the detailed comments, which we address individually below.
R2.2	20. Overall, the authors claim to have proposed a novel method for assessing bridge risk on a global scale. However, as I understand it, the authors have integrated potentially usable remote sensing technology, such as InSAR, into an existing assessment framework for bridge risk assessment, where the ability to be monitored plays an important role. It is well-known that InSAR is a globally usable technology, though it may be limited by certain issues. The authors analyze comparison results before and after incorporating InSAR measurements for global bridge risk assessment, noting that risk accuracy/reliability improves with this integration. However, after reviewing this manuscript multiple times, it remains unclear what specific types of risks the authors analyze and how, as they only highlight "multi-hazard" throughout the title, content, and results. Although the landslide, subsidence, and structure deformation are highlighted.	We thank the Reviewer for the thoughtful feedback. To eliminate ambiguity and more accurately reflect the scope of our study, we have revised the manuscript to replace all references to "multi-hazard" with "geo-hazard," including in the title. The introduction has also been updated to better articulate the research gap, objectives, and the specific types of hazards considered, namely landslides and subsidence. We also emphasize that this work provides a novel quantification of the added value of InSAR monitoring in a large-scale, geo-hazard-oriented bridge risk framework, an aspect not previously addressed in the literature. Finally, we have clarified limitations of the MT-InSAR with the following addition to the discussion: "Additionally, MT-InSAR has inherent limitations requiring expert interpretation to distinguish expected movements (e.g., thermal dilation)

	Additionally, it is unclear whether bridge deformation is measurable, even with many PS measurements on the bridge, because bridge deformation differs significantly from other hazards. Even if deformation is captured, it may not be of concern to engineers, such as thermal dilation or vibration. I hope the authors can redesign the paper to eliminate these ambiguities.	from potentially alarming displacements (e.g., geo-hazard induced). MT-InSAR is also unable to measure rapid displacements exceeding phase unwrapping thresholds or movements occurring primarily in the north-south direction. Therefore, even dense PS coverage does not guarantee detection of all signs of structural degradation." (Lines 642-650)
R2.3	My other comments, as shown below, 1. According to my reading, the manuscript could benefit from a clearer logical flow. Some parts that belong in the discussion section appear in the results section, and the transitions between sections need better connectivity.	We appreciate the Reviewer's observation. In response, we have revised the Results section to improve its clarity and logical flow. Content more appropriate for the Discussion from the "PS availability and spaceborne monitoring" and "PS availability and bridge physical properties" result subsections has been relocated accordingly, and transitions between sections have been refined to ensure a clearer and more coherent narrative throughout the manuscript.
R2.4	2. After I finished browsing through the entire paper, I noted the length of the bridge investigated is beyond 150 m. In my opinion, it would be better to indicate the specific length of the bridge explored at the beginning of this paper. So that the readers can know more information without question in advance.	The length of the bridges in the database has been included in the Introduction in the following sentence: "Applied to a global dataset of 744 long-span bridges, i.e. those with main spans exceeding 150 meters, our methodology employs open-source subsidence and landslide hazards global datasets, calculates exposure metrics based on bridge functional characteristics, and combines these with the enhanced vulnerability factors to generate comprehensive risk profiles that inform monitoring decisions." (Lines 199-206)
R2.5	3. The title may not accurately reflect the content, which is overdefined, as it does not fully address multi-hazard risks and focuses solely on InSAR measurement availability rather than all remote sensing technologies.	The title has been updated to better reflect paper's content and is now as follows: "Global Geo-hazard Risk Assessment of Long-Span Bridges Enhanced with InSAR Availability"
R2.6	3. In lines 48 to 51, it would be better to consider revising the statement following the introduction of vulnerability for better clarity and flow.	The sentence has been moved to the end of this paragraph to improve flow. (Lines 60-63)

R2.7	4. While Sentinel SAR technology offers regular monitoring, it does not provide real-time data. Avoid overstating its capability for near-real-time monitoring, especially in structural health monitoring contexts.	Reference to near-real time has been replaced with "frequent acquisitions". (Line 91)
R2.8	5. In Line-97 to 102, I don't understand the causal relationship in this paragraph. Is the author trying to emphasize their previous work on predicting PS points in the reference of 47?	This paper builds on our previous work on predicting PS points as the method proposed in reference 47 was used to predict availability of MT-InSAR monitoring over bridges. The paragraph has been updated to clearer present the casual relationship between described concepts, and is now as follows: "The European satellite Sentinel-1 provides free global SAR coverage, theoretically enabling worldwide structural monitoring. However, some bridges may not interact effectively with radar wavelengths, limiting MT-InSAR's applicability. Previously, assessing the feasibility of a structure's spaceborne monitoring required data preprocessing, but recent advances enable the prediction of MT-InSAR PS point availability⁴⁷, allowing for assessment of MT-InSAR usability for bridge monitoring prior to data acquisition." (Lines 103-113)
R2.9	6. In Line-122 to 126, the claim that remote sensing has not been used to reduce in-person surveys in bridge risk assessment is inaccurate. Many studies have explored bridge deformation using InSAR. Additionally, this study does not fully address this issue. Could you please identify the real gap for this study?	The introduction has been updated, including a clearer definition of the research gap: "Therefore, the research gap in MT-InSAR-based monitoring of geo-hazard-affected bridges is twofold. Firstly, whilst MT-InSAR is valuable for monitoring such bridges, PS availability is not guaranteed and could limit wide-scale application. We therefore employ recent advances in PS prediction to quantify MT-InSAR's potential for bridge monitoring. Secondly, geo-hazard-affected bridges exhibit time-variant vulnerability through localised dynamic changes that traditional inspections may miss. SHM and MT-InSAR provide more frequent status updates than visual inspections, reducing uncertainties. We thus propose integrating monitoring system availability into geo-hazard risk assessments of bridges." (Lines 170-187)

R2.10	7. In line 141, the authors claim that they already evaluate the subsidence and landslide hazards using global datasets. The term "evaluates" may not be appropriate, as the study integrates multi-hazard results from open sources and InSAR availability rather than conducting original evaluations.	The sentence has been updated and is now as follows: "Applied to a global dataset of 744 long-span bridges, i.e. those with main spans exceeding 150 meters, our methodology employs open-source subsidence and landslide hazards global datasets, calculates exposure metrics based on bridge functional characteristics, and combines these with the enhanced vulnerability factors to generate comprehensive risk profiles that inform monitoring decisions." (Lines 199-206)
R2.11	8. In line 175 to 178, there is a discrepancy between the number of indices mentioned in the text (three) and those in Table 1 (two). Please clarify.	We appreciate the Reviewer pointing out this inconsistency. The sentence was indeed incorrect, so it has been updated and now reads: "Each bridge was assigned a PS availability class based on two predicted metrics: average PS density per 100 meters and the proportion of the bridge covered by PS, following Table 1." (Lines 236-240)
R2.12	9. The placement of the section on factors affecting PS availability seems awkward. Consider moving it to the discussion section.	We agree with this observation. To improve structure and readability, parts of this section have been moved to the Discussion section.
R2.13	10. In line-210, the claim that there is no relationship between bridge material and PS availability needs further discussion or supporting references. Because it directly affects your findings and conclusions.	We have extended the discussion to clarify the relationship between bridge material and PS availability: "Correlations between bridge PS availability and bridge characteristics were analysed. Whilst the physical properties of materials can impact the strength of the SAR signal return, no clear correlation between the bridge material and its potential for monitoring was found. This could be attributed to the fact that bridges are typically constructed from steel and concrete, both of which are characterised by good reflective properties, or it might be because the radar signal is primarily reflected by the superstructure or additional equipment installed on the bridge, such as signs or lampposts." (Lines 439-441)

R2.14	11. In line-229 to 238, please provide more explanation on why bridges-oriented E-W have higher PS availability, as this conflicts with common understanding, especially for certain bridge types. Also, clarify why azimuth does not significantly affect PS availability. So, could you please provide some detailed information?	We appreciate the reviewer's insight. While it is true that bridges oriented along N-S are seen by the satellite from the side, potentially allowing for numerous reflections from the structure's profile, this does not necessarily prevent satellites from effectively observing E-W oriented bridges. As mentioned in the response above, the backscatter signal can be generated by various structural elements above the deck. Additionally, Sentinel-1's spatial resolution in Interferometric Wide swath mode is approximately 5 m in the range direction (roughly E-W) and 20 m in the azimuth direction (roughly N-S). This asymmetric resolution means that E-W oriented bridges are sampled by more pixels along their length, potentially providing more opportunities for persistent scatterer identification. We have extended the discussion to clarify these aspects: "The bridge's azimuth did not significantly affect PS availability. Although previous literature demonstrated that linear infrastructure, such as roads, has better reflectivity when perpendicular to the satellite orbit³⁴, the analysis performed in this study showed that this effect is not pronounced for long-span bridges. This may be attributed to Sentinel-1's asymmetric spatial resolution, which is approximately 5 m in the range direction (roughly E-W) and 20 m in the azimuth direction (roughly N-S). Consequently, E-W oriented bridges are sampled by more pixels along their length compared to N-S oriented structures, potentially providing more opportunities for PS detection. While E-W-oriented bridges are favourable, observing bridges with different azimuths is also feasible." (Lines 456-469)
R2.15	12. In Figure 1, please provide more context for terms like "s/c" in Figures 1 and 7a to avoid confusion for readers from other disciplines.	All references to "s/c" have been updated to "satellite"
R2.16	13. For the results in Figure 3a, Figure 4a, Figure 5a and Figure 6, in my opinion, it is improper to use a small number of the bridge from the region like Africa and Oceania as the representative. It may lead to biased conclusions; especially the reduction percentage of risk level may have some error and be biased. Even you have achieved the bridge information from	We thank the reviewer for highlighting this important point. While we made extensive efforts to identify additional datasets, comprehensive global bridge databases with the level of detail required for this analysis remain limited. We have updated the discussion to acknowledge this constraint:

	the open-source data, could you please spend more time collecting more information about them? The corresponding experiment should be updated or redesigned.	"Due to data collection challenges encountered by the authors of the bridge database³⁰, certain regions, particularly Africa and Oceania, are represented by fewer bridges in the dataset. This spatial imbalance potentially introduces bias in regional statistics, which should therefore be interpreted with caution. [...] These limitations, i.e., spatial imbalance in the bridge database and coarse resolution of data underlying exposure and vulnerability evaluation, reflect the inherent trade-off in conducting global-scale infrastructure analysis, where comprehensive spatial coverage must be balanced against data granularity and completeness. Nevertheless, the methodology presented here provides a valuable framework that can be readily applied to more comprehensive regional datasets as they become available." (Lines 618-623, 631-639)
R2.17	14. The section title of "including monitoring in the risk assessment" clarifies what type of monitoring is included in the risk assessment section.	The section title has been updated to "Including SHM and spaceborne monitoring availability in the risk assessment".
R2.18	15. In line 338 to 350, consider moving the discussion on PS availability to the discussion section where it is addressed, which has been stated in the previous questions.	We appreciate the suggestion and have moved the discussion of PS availability to the discussion section.
R2.19	16. In line 394, could you please explain why vertical deformation is more critical for bridge structural health? Also, I have checked some papers indicating that the bridge deformation can reach up to 1 meter as well as the lateral displacement. On the other hand, for such a kind of deformation, can the InSAR truly capture it? In my opinion, like we highlighted before, the author should be careful to introduce or clarify what kind of bridge deformation you are investigating.	The sentence in former line 394 was indeed misleading and has been clarified as follows: "Nevertheless, spaceborne measurements would still be capable of detecting vertical deformations on such bridges, which could provide some insights into structural health." (Lines 474-478) We have also extended the discussion to include limitations of MT-InSAR: "Additionally, MT-InSAR has inherent limitations requiring expert interpretation to distinguish expected movements (e.g., thermal dilation) from potentially alarming displacements (e.g., geo-hazard induced). MT-InSAR is also unable to measure rapid displacements exceeding phase unwrapping thresholds or movements occurring primarily in the north-south

		direction. Therefore, even dense PS coverage does not guarantee detection of all signs of structural degradation." (Lines 642-650)
R2.20	17. In the section on hazards, describe how you integrated hazard information for different types, like landslides and subsidence, given their varying spatial resolutions.	The section describing hazard methodology has been updated to clearly describe the integration: "After obtaining the maximum subsidence and landslide normalised hazard levels, they were combined into a single geo-hazard value per structure using a geometric mean calculation with inversion and re-scaling procedures, as recommended by the INFORM framework⁷⁴, to ensure higher values correspond to greater hazard severity. The detailed methodology for this geometric mean calculation is described further in this manuscript in the risk calculation subsection." (Lines 814-822)
R2.21	18. Include a new section detailing the open-source datasets used in your study.	The methodology flowchart has been updated to include the list of all datasets used. Additionally, the Data Availability section now includes information about all the datasets.
R2.22	19. In line 762, define "per pixel" and clarify the type of pixel. Specify if you analyzed PS availability over a buffer zone centered on the bridge.	The paragraph has been extended to clarify pixel definition and pixel selection for PS availability analysis and it now reads as follows: "Next, the number of PSs per pixel was estimated using the method proposed in ⁴⁷. This involved downloading long-term coherence data from ⁷⁸ for summer (Northern Hemisphere) or winter (Southern Hemisphere), which are provided at a spatial resolution of approximately 100×100 metres. Infrastructure maps were then generated using OSM data⁷⁶. These coherence and infrastructure datasets were used as inputs to a pre-trained model to infer pixel-wise PS density for each pixel. This approach estimated the number of PSs for each roughly 100x100-meter pixel that intersected with the bridge geometry, without applying additional buffer zones around the bridge structure." (Lines 954-965)

ID	Reviewer #3 (Remarks to the Author):	Response
R3.1	The manuscript presents a novel, globally scalable methodology for multi-hazard risk assessment of long-span bridges, integrating remote sensing (MT-InSAR) availability into structural vulnerability estimation. The work is timely and relevant, offering valuable insights into how satellite-based monitoring can supplement traditional Structural Health Monitoring (SHM) systems, particularly in underserved regions. The manuscript is generally well written and scientifically sound. However, I believe that several critical methodological aspects require clarification or further development before the work can be considered for publication. Major Comments	We appreciate the reviewer's comments and have addressed them point-by-point below.
R3.2	1. Unclear Role of InSAR in Structural Health Monitoring The manuscript does not clearly articulate how spaceborne MT-InSAR can reliably inform on the structural health of bridges, particularly given the fragile failure mechanisms often associated with such structures. It remains ambiguous whether the authors aim to detect long-term displacements due to external factors (e.g., subsidence, landslide) or direct signs of structural deterioration. A more explicit discussion is required on what type of bridge behavior is realistically observable via MT-InSAR, considering its spatial and temporal resolution, and what kind of deterioration mechanisms it can meaningfully capture.	Introduction section has been expanded to explain the potential role of remote sensing in detecting geo-hazard induced displacements: "The impact of localised displacement caused by geo-hazards on bridges' structural health, such as that affecting piers, can be more difficult to detect during periodic inspections compared to the effect of more global displacement that affects the whole structure and often results in more visible functionality issues⁶¹. Thus, remote sensing techniques are particularly valuable for geo-hazard-affected bridges, as they could detect localised deformation patterns that might be overlooked in routine inspections and enable frequent monitoring of progressive movements." (Lines 160-169) Moreover, the discussion section has been updated to clarify MT-InSAR limitations: "Additionally, MT-InSAR has inherent limitations requiring expert interpretation to distinguish expected movements (e.g., thermal dilation) from potentially alarming displacements (e.g., geo-hazard induced). MT-InSAR is also unable to measure rapid displacements exceeding phase unwrapping thresholds or movements occurring primarily in the north-south

		direction. Therefore, even dense PS coverage does not guarantee detection of all signs of structural degradation." (Lines 642-650)
R3.3	2. Limited Hazard Scope and Oversimplified Hazard Integration Only two hazards—subsidence and landslides—are considered, while many regions are predominantly affected by acute hazards such as earthquakes, floods, or hurricanes. This limited hazard spectrum severely restricts the generalizability of the approach. Furthermore, the manuscript treats these two hazards as independent and additive, ignoring potential interdependencies or cascading effects (e.g., subsidence-induced slope failure). This is a substantial limitation in a multi-hazard context and should be more critically discussed. The methodology should be extended or at least conceptually prepared to handle compound risks.	We thank the reviewer for this comment. We acknowledge that our work focuses on two specific hazards and have revised the manuscript to replace "multi-hazard" with "geo-hazard" to better reflect this scope. Additionally, we have extended the introduction to better explain our focus on landslides and subsidence: "Monitoring is crucial in the context of landslides and ground subsidence, which present unique challenges for bridge management due to their progressive nature and the complexity of their structural response. [...] The impact of localised displacement caused by geo-hazards on bridges' structural health, such as that affecting piers, can be more difficult to detect during periodic inspections compared to the effect of more global displacement that affects the whole structure and often results in more visible functionality issues⁶¹. Thus, remote sensing techniques are particularly valuable for geo-hazard-affected bridges, as they could detect localised deformation patterns that might be overlooked in routine inspections and enable frequent monitoring of progressive movements." (Lines 139-142, 160-169) We acknowledge that hazard interdependencies and cascading effects are important factors in understanding infrastructure risk. However, analysing these complex interactions requires a dedicated and in-depth approach, as demonstrated in studies such as https://doi.org/10.1002/2013RG000445 and https://doi.org/10.1038/s41598-023-40400-5. Given the scope of this work, which aligns with established practices in global-scale assessments (e.g. https://doi.org/10.1038/s41467-019-10442-3), we have focused on primary hazards, with the understanding that cascading effects represent a valuable direction for future research. (Lines 599-604)

R3.4	3. Ambiguity in Hazard Calculation The description of how landslide and subsidence hazard levels are incorporated into the risk metric is unclear in the Methods section. It is essential to clarify how these datasets are normalized, weighted, or combined in the final risk computation. Figure 8, intended to summarize the methodology, is overly complex and should be simplified or supplemented with clearer narrative explanations.	Figure 8 has been simplified to better represent the employed methodology. The section on hazards has been updated to clarify the integration.
R3.5	4. Arbitrary Monitoring Reduction Factors The methodology applies vulnerability reduction factors (1.35 to 1.0) based on SHM or InSAR monitoring availability. However, the rationale behind these numerical values is not empirically justified. This approach assumes that the mere presence of monitoring reduces risk, without considering monitoring quality, continuity, or resolution. The authors should explain how these factors were determined and whether any sensitivity analysis was conducted to assess their impact on the results.	We appreciate the reviewer's suggestion. In response we have extended the discussion of the limitations and added a sensitivity analysis highlighting the impact the selection of monitoring factor value has on the final risk metric: "The monitoring factor used in structural vulnerability calculations reflects the assumption that the availability of remotely sensed data allows more frequent structural health updates, thereby reducing uncertainty in the inherently time-variant structural vulnerability and consequently decreasing overall bridge risk. Monitoring improves knowledge about the structure, hence we adopted a coefficient value of 1.35, commonly used in building codes and standards, such as Eurocode-8⁶⁵ and FEMA 356⁶⁶ to account for uncertainties associated with (lack of) knowledge about building materials. Although this is a standardised value, it has not previously been applied in this context. Therefore, to fully understand the impact of this value selection on the final risk score, we performed a sensitivity analysis. Supplementary Figure 5 shows that changes due to different monitoring factor selections would be minor, with few bridges changing their assigned risk category. [...] Moreover, the bridge database used in the study does not specify SHM sensor types, so our assumption of "ideal" monitoring may overestimate its capabilities. Additionally, MT-InSAR has inherent limitations requiring expert interpretation to distinguish expected movements (e.g., thermal dilation) from potentially alarming displacements (e.g., geo-hazard induced). MT-InSAR is also unable to measure rapid displacements exceeding phase unwrapping thresholds or movements occurring primarily in the north-south

		direction. Therefore, even dense PS coverage does not guarantee detection of all signs of structural degradation. Although the monitoring factor represents a simplification, it enables wide-scale estimation of routine remote monitoring benefits, and future studies could expand this factor to account for various SHM sensor types and MT-InSAR limitations." (Lines 563-580, 639-654)
R3.6	5. Use of Static and Coarse-Resolution Input Data Exposure and vulnerability estimations rely on static, low-resolution, or indirectly inferred data (e.g., bridge typologies, OSM-derived road functions, coarse age brackets). The implications of this data coarseness on the reliability of the risk results are not discussed. The authors should elaborate on how these limitations affect the statistical robustness and regional applicability of their method.	We have added this limitation to the discussion: "Furthermore, the exposure and vulnerability assessments necessarily rely on globally available datasets that are inherently static and of relatively coarse resolution. This data limitation has several implications for the risk assessment: regional variations in construction standards and engineering practices may not be adequately captured by broad typological categories, whilst static datasets cannot account for temporal changes in bridge conditions. These limitations, i.e., spatial imbalance in the bridge database and coarse resolution of data underlying exposure and vulnerability evaluation, reflect the inherent trade-off in conducting global-scale infrastructure analysis, where comprehensive spatial coverage must be balanced against data granularity and completeness. Nevertheless, the methodology presented here provides a valuable framework that can be readily applied to more comprehensive regional datasets as they become available." (Lines 623-639)
R3.7	6. Overstated Effectiveness of MT-InSAR Monitoring The paper emphasizes the wide potential of MT-InSAR but does not adequately address its limitations, including:  o Susceptibility to false positives/negatives due to decorrelation and atmospheric effects; o Inability to detect abrupt or high-frequency structural failures (e.g., due to earthquakes); 	In response to this comment and addressing other reviewers concerns we extended the discussion to include a section on MT-InSAR limitations.

	o Oversimplified assumption that PS density correlates directly with effective monitoring, without accounting for structural geometry, orientation (e.g., N-S aligned bridges), or radar-specific limitations. These aspects should be discussed more critically, particularly in the context of decision-making based on such data.	
R3.8	7. Uncertainty Quantification While monitoring is framed as reducing “epistemic uncertainty,” the manuscript does not explain how uncertainty is quantitatively treated or propagated through the risk model. This omission undermines the credibility of the final risk scores and limits their usefulness in prioritization or policy decisions. The authors should at least discuss the current limitations in this regard and propose potential directions for future incorporation of uncertainty modeling.	We thank the reviewer for this valuable observation. While the quantification of uncertainty is indeed an important aspect of risk assessment, it requires comprehensive investigation and is often challenging to assess, lacking widely accepted guidelines (https://doi.org/10.1016/j.ijdr.2025.105260). We acknowledge this as an important constraint of our study and have included it in the discussion: "Additionally, whilst this work assumes that monitoring availability reduces epistemic uncertainties in time-variant structural vulnerability, it does not explicitly quantify uncertainty propagation through the risk model. Future work could therefore investigate more comprehensive quantification of how monitoring availability impacts vulnerability assessment, including the implications of MT-InSAR measurement limitations and potential synergistic effects when both SHM and spaceborne monitoring are available. Such developments would enhance stakeholder confidence in risk scores and improve the robustness of prioritisation decisions." (Lines 580-591)
R3.9	Minor Comments  • Lines 617–618: The manuscript describes landslides as gradual processes. This is misleading, as many landslides—especially those triggered by rainfall or seismic events—can be very rapid. Please revise for accuracy. 	The method section has been updated to clarify this concept: “This article focuses on gradual ground deformation processes, specifically subsidence and slow-moving landslides, that can lead to bridge displacements detectable using MT-InSAR. These hazards typically evolve over time at rates compatible with the temporal sampling frequency of radar acquisitions, allowing for effective monitoring. In contrast, rapid landslides, which involve movement rates that exceed the revisit intervals of radar satellites, fall outside the scope of this study due to their incompatibility with MT-InSAR detection. While this work does not address such rapid events,

		future research could consider their inclusion in broader risk assessments.” (Lines 771-783)
R3.10	 • Line 689: The text states that degradation is classified into five levels, yet Figure S3 seems to depict only four. Please clarify or correct the inconsistency. 	This has been clarified in the Figure S3 and in the main text with the following sentence: “Although the Italian guidelines define five degradation levels, the limited structural health information available in the bridge database allowed for the assignment of only four levels, with the medium-high degradation category remaining unused in this analysis.” (Lines 871-875)
R3.11	 • Line 670 and Figure S3: The description of structural vulnerability should align more closely with the logic and classification levels presented in Figure S3. Greater consistency would improve clarity. 	The description of vulnerability has been modified to match the figure, as follows: “The vulnerability in these guidelines is determined using factors such as the level of degradation, construction period, design code class, and structure's properties.” (Lines 849-853)

REVIEWER COMMENTS

Reviewer #1 (Remarks to the Author):

This reviewer maintains some reservations regarding the feasibility and reliability of the approach related to the arbitrary selection of the monitoring factors (see comment R1.7) and the type of sensor adopted to account for the presence of in-contact monitoring (comment R1.8). However, these issues may be addressed in further research developments. In this reviewer's opinion, the authors' responses have addressed the main issues raised in the original version of the manuscript and it does not require further modifications.

We appreciate your feedback and acknowledge these important considerations for future research direction.

Reviewer #2 (Remarks to the Author):

Dear Authors,

Thank you for your efforts in addressing my comments. It is great to see that you have revised all the figures and included statistical information. The revised manuscript has improved significantly and is much clearer than before. However, there are still some additional points that need to be addressed to avoid confusion for future readers. My comments are listed below:

We appreciate your suggestions and have responded to them point-by-point below.

ID	Comment	Response
R2.1	1. On page-line 67, it is not accurate to say that periodic inspections are insufficient, as the satellite observations used in this study are also a form of periodic inspection. Please clarify the 'real' distinction between traditional inspections and satellite-based monitoring.	We clarified the distinction between the visual and satellite observation by changing this part of the sentence from: "2) standard periodic inspections may be insufficient for timely detection of deterioration" to "2) visual inspections may miss early signs of deterioration or fail to detect changes that

		develop between the typical two-year inspection intervals, making traditional periodic approaches insufficient for timely detection of structural issues”
R2.2	2. On page-line 194, the phrase "reduce global geo-hazard risk" may overstate the impact of this study. In my opinion, this research primarily revises the assessment strategy to potentially improve risk level definitions but does not directly reduce global geo-hazard risks. Please consider rephrasing this to more accurately reflect the contributions.	We have updated the sentence to better reflect this research contribution by changing it from: “This expanded monitoring capability could substantially reduce global geo-hazard risk and decrease the number of bridges classified as high-risk.” to: “Including this expanded monitoring capability in the risk assessment framework refines the risk classification methodology and demonstrates how incorporating spaceborne monitoring can improve risk level definitions and reduce the number of bridges classified as high-risk.”
R2.3	3. Regarding Figure 3a, could you explain why spaceborne monitoring in North America appears to be the least (49%) compared to other regions? Please provide the reason for this observation. Is that due to the Sentinel-1 coverage or not?	This observation requires more detailed investigation to provide a definitive explanation. Sentinel-1 coverage is likely a contributing factor, as North America has numerous long-span bridges concentrated in the eastern part of the country along the Mississippi and Ohio rivers, regions that receive coverage from only one Sentinel orbit rather than two. Additionally, the predictive model used to estimate PS availability may underestimate the actual number of persistent scatterers for North American bridges because the model was trained using ascending orbit coherence data, while the coherence data available for North America comes from descending orbits. This mismatch between training and inference datasets could affect the model's accuracy in predicting PS availability for this region. Further research would be valuable to investigate this issue comprehensively and incorporate orbit-specific training data to improve regional accuracy.
R2.4	4. For Figure 7b, the current description in the text is somewhat difficult to understand. Could you please revise it to make it clearer or consider rephrasing it in a different way?	We have rephrased the description of Figure 7b in the text to improve clarity, so that instead of: “Critically, data from Sentinel-1A were focused in that period of reduced capacity on highly developed regions such as Europe, where the repeat time increased, but data from two flight directions were still available. However, this monitoring schedule left long-span bridges in less privileged areas of the world with significantly reduced monitoring capabilities, including a few bridges in Africa and Latin America that have lost monitoring capabilities entirely (see Figure 7b).”

		it is: “The reduced satellite capacity created regional disparities in monitoring availability. Highly developed regions such as Europe maintained data from two flight directions, although with longer revisit times. However, bridges in less privileged areas experienced significantly reduced capabilities, with some in Africa and Latin America losing spaceborne monitoring entirely (see Figure 7b).”
R2.5	5. On page-line 555, I am unsure if it is appropriate to add a subtitle for the following section, such as "Limitations of this study." However, this is ultimately up to the authors to decide.	We appreciate the suggestion. However, subheadings are not permitted in the discussion section according to the journal's editorial guidelines.
R2.6	6. Regarding Supplementary Figure 3, the authors assigned a higher weight to bridges built after 1980 (which may increase the degradation speed level) and a lower weight to those built before 1945. Could you please provide justification or references to support this weighting scheme? It would be helpful for readers to understand the rationale behind this approach.	Indeed, the weighting scheme assigns higher degradation speed categories to younger bridges (built after 1980) and lower categories to older bridges (built before 1945). This follows directly from the logic established in the Italian guidelines that we adapted. This approach is based on the principle that if a young bridge exhibits heavy degradation, it indicates a high degradation speed, whereas the same level of degradation in an older bridge may simply reflect expected ageing over time (please see references 75 and 77 from the paper). This rationale ensures that bridges experiencing rapid deterioration relative to their age are appropriately identified as higher vulnerability. We extended the figure description to clarify that point: “Logical process for assigning structural vulnerability classes to each bridge, simplified from the Italian guidelines to include only relevant categories. Note that younger bridges are assigned higher degradation speeds following the guidelines' assumption that bridges experiencing rapid deterioration relative to their age are relatively more vulnerable.”

Reviewer #3 (Remarks to the Author):

The authors have satisfactorily addressed all major and minor comments raised in the previous review round. The revised manuscript now presents a clear, well-structured, and methodologically sound study, offering a novel and globally scalable framework for integrating MT-InSAR monitoring potential into geo-hazard risk assessment of long-span bridges.

In detail:

- *The introduction and discussion now explicitly delineate which bridge behaviours can be observed with MT-InSAR, and the limitations related to resolution, displacement direction, and rapid events. This balances enthusiasm for the technique with a realistic appraisal of its applicability.*
- *The terminology has been refined from “multi-hazard” to “geo-hazard,” with a clear justification for focusing on landslides and subsidence, and a well-reasoned explanation of why cascading effects are left for future work.*
- *The Methods section and Figure 8 have been revised for greater clarity, detailing the normalisation and combination of hazard datasets.*
- *The rationale for the 1.35 coefficient is now linked to established engineering codes, and a sensitivity analysis confirms its minimal effect on risk classification.*
- *The discussion now explicitly addresses the coarse resolution and static nature of input datasets, potential regional biases, and implications for interpretation.*
- *Terminology, figure alignment, and classification levels have been harmonised.*

The manuscript reads well, is logically organised, and is accessible to a multidisciplinary audience. Figures are generally clear, although Figure 8, while improved, could still benefit from slightly larger fonts for legibility in print. I did not identify any remaining substantive methodological issues or inconsistencies. Accept in present form, subject only to minor editorial polishing by the journal.

We appreciate your feedback and are pleased that our revisions have successfully addressed all the points you raised. We have increased the font and overall size of Figure 8 as suggested to enhance readability. We're grateful for your recommendation to accept the manuscript and look forward to its publication.